# Learning Energy-Based Prior Model with Diffusion-Amortized MCMC

**Peiyu Yu**[1]
yupeiyu98@g.ucla.edu

**Yaxuan Zhu**[1]
yaxuanzhu@g.ucla.edu

**Sirui Xie**[2]
srxie@ucla.edu

**Xiaojian Ma**[2,4]
xiaojian.ma@ucla.edu

**Ruiqi Gao**[3]
ruiqig@google.com

**Song-Chun Zhu**[4]
sczhu@stat.ucla.edu

**Ying Nian Wu**[1]
ywu@stat.ucla.edu

[1]UCLA Department of Statistics     [2]UCLA Department of Computer Science
[3]Google DeepMind [4]Beijing Institute for General Artificial Intelligence (BIGAI)

## Abstract

Latent space Energy-Based Models (EBMs), also known as energy-based priors, have drawn growing interests in the field of generative modeling due to its flexibility in the formulation and strong modeling power of the latent space. However, the common practice of learning latent space EBMs with non-convergent short-run MCMC for prior and posterior sampling is hindering the model from further progress; the degenerate MCMC sampling quality in practice often leads to degraded generation quality and instability in training, especially with highly multimodal and/or high-dimensional target distributions. To remedy this sampling issue, in this paper we introduce a simple but effective diffusion-based amortization method for long-run MCMC sampling and develop a novel learning algorithm for the latent space EBM based on it. We provide theoretical evidence that the learned amortization of MCMC is a valid long-run MCMC sampler. Experiments on several image modeling benchmark datasets demonstrate the superior performance of our method compared with strong counterparts[1].

## 1 Introduction

Generative modeling of data distributions has achieved impressive progress with the fast development of deep generative models in recent years [1–9]. It provides a powerful framework that allows successful applications in synthesizing data of different modalities [10–15], extracting semantically meaningful data representation [16–18] as well as other important domains of unsupervised or semi-supervised learning [19–21]. A fundamental and powerful branch of generative modeling is the Deep Latent Variable Model (DLVM). Typically, DLVM assumes that the observation (*e.g.*, a piece of text or images) is generated by its corresponding low-dimensional latent variables via a top-down generator network [1–3]. The latent variables are often assumed to follow a non-informative prior distribution, such as a uniform or isotropic Gaussian distribution. While one can directly learn a deep top-down generator network to faithfully map the non-informative prior distribution to the data distribution, learning an informative prior model in the latent space could further improve the ex-

---

[1]Code and data available at https://github.com/yuPeiyu98/Diffusion-Amortized-MCMC.

37th Conference on Neural Information Processing Systems (NeurIPS 2023).

pressive power of the DLVM with significantly less parameters [22]. In this paper, we specifically consider learning an EBM in the latent space as an informative prior for the model.

Learning energy-based prior can be challenging, as it typically requires computationally expensive Markov Chain Monte Carlo (MCMC) sampling to estimate learning gradients. The difficulty of MCMC-based sampling is non-negligible when the target distribution is highly multi-modal or high-dimensional. In these situations, MCMC sampling can take a long time to converge and perform poorly on traversing modes with limited iterations [23]. Consequently, training models with samples from non-convergent short-run MCMC [24], which is a common choice for learning latent space EBMs [22], often results in malformed energy landscapes [15, 24, 25] and biased estimation of the model parameter. One possible solution is to follow the variational learning scheme [1], which however requires non-trivial extra efforts on model design to deal with problems like posterior collapse [26–28] and limited expressivity induced by model assumptions [1, 29, 30].

To remedy this sampling issue and further unleash the expressive power of the prior model, we propose to shift attention to an economical compromise between unrealistically expensive long-run MCMC and biased short-run MCMC: *we consider learning valid amortization of the potentially long-run MCMC for learning energy-based priors*. Specifically, inspired by the connection between MCMC sampling and denoising diffusion process [7, 8, 31], in this paper we propose a diffusion-based amortization method suitable for long-run MCMC sampling in learning latent space EBMs. The learning algorithm derived from it breaks the long-run chain into consecutive affordable short-run segments that can be iteratively distilled by a diffusion-based sampler. The core idea is simple and can be summarized by a one-liner (Fig. 1). We provide theoretical and empirical evidence that the resulting sampler approximates the long-run chain (see the proof-of-concept toy examples in Appendix E.1), and brings significant performance improvement for learning latent space EBMs on several tasks. We believe that this proposal is a notable attempt to address the learning issues of energy-based priors and is new to the best of our knowledge. We kindly refer to Section 5 for a comprehensive discussion of the related work.

**Contributions**   i) We propose a diffusion-based amortization method for MCMC sampling and develop a novel learning algorithm for the latent space EBM. ii) We provide a theoretical understanding that the learned amortization of MCMC is a valid long-run MCMC sampler. iii) Our experiments demonstrate empirically that the proposed method brings higher sampling quality, a better-learned model and stronger performance on several image modeling benchmark datasets.

## 2   Background

### 2.1   Energy-Based Prior Model

We assume that for the observed sample $\boldsymbol{x} \in \mathbb{R}^D$, there exists $\boldsymbol{z} \in \mathbb{R}^d$ as its unobserved latent variable vector. The complete-data distribution is

$$p_{\boldsymbol{\theta}}(\boldsymbol{z}, \boldsymbol{x}) := p_{\boldsymbol{\alpha}}(\boldsymbol{z}) p_{\boldsymbol{\beta}}(\boldsymbol{x}|\boldsymbol{z}), \; p_{\boldsymbol{\alpha}}(\boldsymbol{z}) := \frac{1}{Z_{\boldsymbol{\alpha}}} \exp\left(f_{\boldsymbol{\alpha}}(\boldsymbol{z})\right) p_0(\boldsymbol{z}), \qquad (1)$$

where $p_{\boldsymbol{\alpha}}(\boldsymbol{z})$ is the prior model with parameters $\boldsymbol{\alpha}$, $p_{\boldsymbol{\beta}}(\boldsymbol{x}|\boldsymbol{z})$ is the top-down generation model with parameters $\boldsymbol{\beta}$, and $\boldsymbol{\theta} = (\boldsymbol{\alpha}, \boldsymbol{\beta})$. The prior model $p_{\boldsymbol{\alpha}}(\boldsymbol{z})$ can be formulated as an energy-based model, which we refer to as the Latent-space Energy-Based Model (LEBM) [22] throughout the paper. In this formulation, $f_{\boldsymbol{\alpha}}(\boldsymbol{z})$ is parameterized by a neural network with scalar output, $Z_{\boldsymbol{\alpha}}$ is the partition function, and $p_0(\boldsymbol{z})$ is standard normal as a reference distribution. The prior model in Eq. (1) can be interpreted as an energy-based correction or exponential tilting of the original prior distribution $p_0$. The generation model follows $p_{\boldsymbol{\beta}}(\boldsymbol{x}|\boldsymbol{z}) = \mathcal{N}(g_{\boldsymbol{\beta}}(\boldsymbol{z}), \sigma^2 \mathbf{I}_D)$, where $g_{\boldsymbol{\beta}}$ is the generator network and $\sigma^2$ takes a pre-specified value as in VAE [1]. This is equivalent to using $l_2$ error for reconstruction.

The parameters of LEBM and the generation model can be learned by Maximum Likelihood Estimation (MLE) [22]. To be specific, given the training data $\boldsymbol{x}$, the gradients for updating $\boldsymbol{\alpha}, \boldsymbol{\beta}$ are,

$$\delta_{\boldsymbol{\alpha}}(\boldsymbol{x}) := \mathbb{E}_{p_{\boldsymbol{\theta}}(\boldsymbol{z}|\boldsymbol{x})}\left[\nabla_{\boldsymbol{\alpha}} f_{\boldsymbol{\alpha}}(\boldsymbol{z})\right] - \mathbb{E}_{p_{\boldsymbol{\alpha}}(\boldsymbol{z})}\left[\nabla_{\boldsymbol{\alpha}} f_{\boldsymbol{\alpha}}(\boldsymbol{z})\right], \; \delta_{\boldsymbol{\beta}}(\boldsymbol{x}) := \mathbb{E}_{p_{\boldsymbol{\theta}}(\boldsymbol{z}|\boldsymbol{x})}\left[\nabla_{\boldsymbol{\beta}} \log p_{\boldsymbol{\beta}}(\boldsymbol{x}|\boldsymbol{z})\right]. \; (2)$$

In practice, one may use the Monte-Carlo average to estimate the expectations in Eq. (2). This involves sampling from the prior $p_{\boldsymbol{\alpha}}(\boldsymbol{z})$ and the posterior $p_{\boldsymbol{\theta}}(\boldsymbol{z}|\boldsymbol{x})$ distribution using MCMC, specifically Langevin Dynamics (LD) [32], to estimate the expectations and hence the gradient. For a target

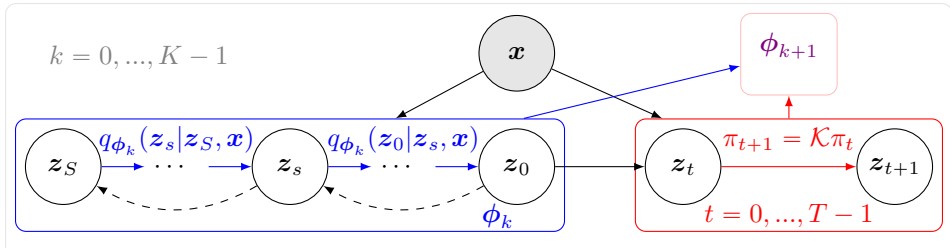

Figure 1: **Learning the DAMC sampler.** The training samples for updating the sampler to $\phi_{k+1}$ is obtained by $T$-step short-run LD, initialized with the samples from the current learned sampler $\phi_k$. Best viewed in color.

distribution $\pi(\boldsymbol{z})$, the dynamics iterates

$$\boldsymbol{z}_{t+1} = \boldsymbol{z}_t + \frac{s^2}{2}\nabla_{\boldsymbol{z}_t}\log\pi(\boldsymbol{z}_t) + s\boldsymbol{w}_t, \ t = 0, 1, ..., T-1, \ \boldsymbol{w}_t \sim \mathcal{N}(\mathbf{0}, \mathbf{I}_d), \tag{3}$$

where $s$ is a small step size. One can draw $\boldsymbol{z}_0 \sim \mathcal{N}(\mathbf{0}, \mathbf{I}_d)$ to initialize the chain. For sufficiently small step size $s$, the distribution of $\boldsymbol{z}_t$ will converge to $\pi$ as $t \to \infty$ [32]. However, it is prohibitively expensive to run LD until convergence in most cases, for which we may resort to limited iterations of LD for sampling in practice. This non-convergent short chain yields a moment-matching distribution close to the true $\pi(\boldsymbol{z})$ but is often biased, which was dubbed as short-run LD [15, 23–25].

## 2.2 Denoising Diffusion Probabilistic Model

Closely related to EBMs are the Denoising Diffusion Probabilistic Models (DDPMs) [5, 7, 8]. As pointed out in [5, 8], the sampling procedure of DDPM with $\epsilon$-prediction parametrization resembles LD; $\epsilon$ (predicted noise) plays a similar role to the gradient of the log density [8].

In the formulation proposed by Kingma et al. [33], the DDPM parameterized by $\phi$ is specified by a noise schedule built upon $\lambda_s = \log[\beta_s^2/\sigma_s^2]$, *i.e.*, the log signal-to-noise-ratio, that decreases monotonically with $s$. $\beta_s$ and $\sigma_s^2$ are strictly positive scalar-valued functions of $s$. We use $\boldsymbol{z}_0$ to denote training data in $\mathbb{R}^d$. The forward-time diffusion process $q(\boldsymbol{z}|\boldsymbol{z}_0)$ is defined as:

$$q(\boldsymbol{z}_s|\boldsymbol{z}_0) = \mathcal{N}(\boldsymbol{z}_s; \beta_s\boldsymbol{z}_0, \sigma_s^2\mathbf{I}_d), \quad q(\boldsymbol{z}_s'|\boldsymbol{z}_s) = \mathcal{N}(\boldsymbol{z}_s'; (\beta_{s'}/\beta_s)\boldsymbol{z}_s, \sigma_{s'|s}^2\mathbf{I}_d), \tag{4}$$

where $0 \leqslant s < s' \leqslant S$ and $\sigma_{s'|s}^2 = (1 - e^{\lambda_{s'}-\lambda_s})\sigma_s^2$. Noticing that the forward process can be reverted as $q(\boldsymbol{z}_s|\boldsymbol{z}_{s'}, \boldsymbol{z}_0) = \mathcal{N}(\boldsymbol{z}_s; \tilde{\boldsymbol{\mu}}_{s|s'}(\boldsymbol{z}_{s'}, \boldsymbol{z}_0), \tilde{\sigma}_{s|s'}^2\mathbf{I}_d)$, an ancestral sampler $q_\phi(\boldsymbol{z}_s|\boldsymbol{z}_{s'})$ [8] that starts at $\boldsymbol{z}_S \sim \mathcal{N}(\mathbf{0}, \mathbf{I}_d)$ can be derived accordingly [33]:

$$\tilde{\boldsymbol{\mu}}_{s|s'}(\boldsymbol{z}_{s'}, \boldsymbol{z}_0) = e^{\lambda_{s'}-\lambda_s}(\alpha_s/\alpha_{s'})\boldsymbol{z}_{s'} + (1 - e^{\lambda_{s'}-\lambda_s})\alpha_s\boldsymbol{z}_0, \quad \tilde{\sigma}_{s|s'}^2 = (1 - e^{\lambda_{s'}-\lambda_s})\sigma_s^2,$$
$$\boldsymbol{z}_s = \tilde{\boldsymbol{\mu}}_{s|s'}(\boldsymbol{z}_{s'}, \hat{\boldsymbol{z}}_0) + \sqrt{(\tilde{\sigma}_{s|s'}^2)^{1-\gamma}(\sigma_{s'|s}^2)^\gamma}\boldsymbol{\epsilon}, \tag{5}$$

where $\boldsymbol{\epsilon}$ is standard Gaussian noise, $\hat{\boldsymbol{z}}_0$ is the prediction of $\boldsymbol{z}_0$ by the DDPM $\phi$, and $\gamma$ is a hyperparameter that controls the noise magnitude, following [34]. The goal of DDPM is to recover the distribution of $\boldsymbol{z}_0$ from the given Gaussian noise distribution. It can be trained by optimizing $\mathbb{E}_{\boldsymbol{\epsilon},\lambda}\left[\|\boldsymbol{\epsilon}(\boldsymbol{z}_\lambda) - \boldsymbol{\epsilon}\|_2^2\right]$, where $\boldsymbol{\epsilon} \sim \mathcal{N}(\mathbf{0}, \mathbf{I}_d)$ and $\lambda$ is drawn from a distribution of log noise-to-signal ratio $p(\lambda)$ over uniformly sampled times $s \in [0, S]$. This loss can be justified as a lower bound on the data log-likelihood [8, 33] or as a variant of denoising score matching [7, 35]. We will exploit in this paper the connection between DDPMs and LD sampling of EBMs, based upon which we achieve better sampling performance for LEBM compared with short-run LD.

## 3 Method

In this section, we introduce the diffusion-based amortization method for long-run MCMC sampling in learning LEBM in Section 3.1. The learning algorithm of LEBM based on the proposed method and details about implementation are then presented in Section 3.2 and Section 3.3, respectively.

## 3.1 Amortizing MCMC with DDPM

**Amortized MCMC**  Using the same notation as in Section 2, we denote the starting distribution of LD as $\pi_0(z)$, the distribution after $t$-th iteration as $\pi_t(z)$ and the target distribution as $\pi(z)$. The trajectory of LD in Eq. (3) is typically specified by its transition kernel $\mathcal{K}(z|z')$. The process starts with drawing $z_0$ from $\pi_0(z)$ and iteratively sample $z_t$ at the $t$-th iteration from the transition kernel conditioned on $z_{t-1}$, *i.e.*, $\pi_t(z) = \mathcal{K}\pi_{t-1}(z)$, where $\mathcal{K}\pi_{t-1}(z) := \int \mathcal{K}(z|z')\pi_{t-1}(z')dz'$. Recursively, $\pi_t = \mathcal{K}_t\pi_0$, where $\mathcal{K}_t$ denotes the $t$-step transition kernel. LD can therefore be viewed as approximating a fixed point update in a non-parametric fashion since the target distribution $\pi$ is a stationary distribution $\pi(z) := \int \mathcal{K}(z|z')\pi(z')dz', \forall z$. This motivates several works for more general approximations of this update [36–38] with the help of neural samplers.

Inspired by [36], we propose to use the following framework for amortizing the LD in learning the LEBM. Formally, let $\mathcal{Q} = \{q_\phi\}$ be the set of amortized samplers parameterized by $\phi$. Given the transition kernel $\mathcal{K}$, the goal is to find a sampler $q_{\phi*}$ to closely approximate the target distribution $\pi$. This can be achieved by iteratively distill $T$-step transitions of LD into $q_\phi$:

$$q_{\phi_k} \leftarrow \arg\min_{q_\phi \in \mathcal{Q}} \mathcal{D}[q_{\phi_{k-1},T}||q_\phi], \ q_{\phi_{k-1},T} := \mathcal{K}_T q_{\phi_{k-1}}, \ q_{\phi_0} \approx \pi_0, \ k = 0, ..., K-1. \quad (6)$$

where $\mathcal{D}[\cdot||\cdot]$ is the Kullbeck-Leibler Divergence (KLD) measure between distributions. Concretely, Eq. (6) means that to recover the target distribution $\pi$, instead of using long-run LD, we can repeat the following steps: i) employ a $T$-step short-run LD initialized with the current sampler $q_{\phi_{k-1}}$ to approximate $\mathcal{K}_T q_{\phi_{k-1}}$ as the target distribution of the current sampler, and ii) update the current sampler $q_{\phi_{k-1}}$ to $q_{\phi_k}$. The correct convergence of $q_\phi$ to $\pi$ with Eq. (6) is supported by the standard theory of Markov chains [39], which suggests that the update in Eq. (6) is monotonically decreasing in terms of KLD, $\mathcal{D}[q_{\phi_k}||\pi] \leqslant \mathcal{D}[q_{\phi_{k-1}}||\pi]$. We refer to Appendix A.1 for a detailed explanation and discussion of this statement. In practice, one can apply gradient-based methods to minimize $\mathcal{D}[q_{\phi_{k-1},T}||q_\phi]$ and approximate Eq. (6) for the update from $q_{\phi_{k-1}}$ to $q_{\phi_k}$. The above formulation provides a generic and flexible framework for amortizing the potentially long MCMC.

**Diffusion-based amortization**  To avoid clutter, we simply write $q_{\phi_{k-1},T}$ as $q_T$. We can see that

$$\arg\min_{q_\phi} \mathcal{D}[q_T||q_\phi] = \arg\min_{q_\phi} -\mathcal{H}(q_T) + \mathcal{H}(q_T, q_\phi) = \arg\min_{q_\phi} -\mathbb{E}_{q_T}[\log q_\phi], \quad (7)$$

where $\mathcal{H}$ represents the entropy of distributions. The selection of the sampler $q_\phi$ is a matter of design. According to Eq. (7), we may expect the following properties from $q_\phi$: i) having analytically tractable expression of the exact value or lower bound of log-likelihood, ii) easy to draw samples from and iii) capable of close approximation to the given distribution $\{q_T\}$. In practice, iii) is important for the convergence of Eq. (6). If $q_\phi$ is far away from $q_T$ in each iteration, then non-increasing KLD property $\mathcal{D}[q_{\phi_k}||\pi] \leqslant \mathcal{D}[q_{\phi_{k-1}}||\pi]$ might not hold, and the resulting amortized sampler would not converge to the true target distribution $\pi(z)$.

For the choice of $q_\phi$, let us consider distilling the gradient field of $q_T$ in each iteration, so that the resulting sampler is close to the $q_T$ distribution. This naturally points to the DDPMs [8]. To be specific, learning a DDPM with $\epsilon$-prediction parameterization is equivalent to fitting the finite-time marginal of a sampling chain resembling annealed Langevin dynamics [7, 8, 31]. Moreover, it also fulfills i) and ii) of the desired properties mentioned above. We can plug in the objective of DDPM (Section 2.2), which is a lower bound of $\log q_\phi$, to obtain the gradient-based update rule for $q_\phi$:

$$\phi_{k-1}^{(i+1)} \leftarrow \phi_{k-1}^{(i)} - \eta \nabla_\phi \mathbb{E}_{\epsilon,\lambda}\left[\|\epsilon(z_\lambda) - \epsilon\|_2^2\right], \ \phi_k^{(0)} \leftarrow \phi_{k-1}^{(M)}, \ i = 0, 1, ..., M-1 \quad (8)$$

where $\epsilon \sim \mathcal{N}(0, \mathbf{I}_d)$. $\lambda$ is drawn from a distribution of log noise-to-signal ratio $p(\lambda)$. $\eta$ is the step size for the update, and $M$ is the number of iterations needed in Eq. (8). In practice, we find that when amortizing the LD sampling chain, a light-weight DDPM $q_\phi$ updated with very small $M$, *i.e.*, few steps of Eq. (8) iteration, approximates Eq. (6) well. We provide a possible explanation using the Fisher information by scoping the asymptotic behavior of this update rule in the Appendix A.2. We term the resulting sampler as Diffusion-Amortized MCMC (DAMC).

## 3.2 Approximate MLE with DAMC

In this section, we show how to integrate the DAMC sampler into the learning framework of LEBM and form a symbiosis between these models. Given a set of $N$ training samples $\{x_i\}_{i=1}^N$ independently drawn from the unknown data distribution $p_{\text{data}}(x)$, the model $p_\theta$ (Section 2.1) can be trained

by maximizing the log-likelihood over training samples $\mathcal{L}(\boldsymbol{\theta}) = \frac{1}{N}\sum_{i=1}^{N} \log p_{\boldsymbol{\theta}}(\boldsymbol{x}_i)$. Doing so typically requires computing the gradients of $\mathcal{L}(\boldsymbol{\theta})$, where for each $\boldsymbol{x}_i$ the learning gradient satisfies:

$$
\begin{aligned}
\nabla_{\boldsymbol{\theta}} \log p_{\boldsymbol{\theta}}(\boldsymbol{x}_i) &= \mathbb{E}_{p_{\boldsymbol{\theta}}(\boldsymbol{z}_i|\boldsymbol{x}_i)}\left[\nabla_{\boldsymbol{\theta}} \log p_{\boldsymbol{\theta}}(\boldsymbol{z}_i, \boldsymbol{x}_i)\right] \\
&= (\underbrace{\mathbb{E}_{p_{\boldsymbol{\theta}}(\boldsymbol{z}_i|\boldsymbol{x}_i)}\left[\nabla_{\boldsymbol{\alpha}} f_{\boldsymbol{\alpha}}(\boldsymbol{z}_i)\right] - \mathbb{E}_{p_{\boldsymbol{\alpha}}(\boldsymbol{z}_i)}\left[\nabla_{\boldsymbol{\alpha}} f_{\boldsymbol{\alpha}}(\boldsymbol{z}_i)\right]}_{\delta_{\boldsymbol{\alpha}}(\boldsymbol{x}_i)}, \underbrace{\mathbb{E}_{p_{\boldsymbol{\theta}}(\boldsymbol{z}_i|\boldsymbol{x}_i)}\left[\nabla_{\boldsymbol{\beta}} \log p_{\boldsymbol{\beta}}(\boldsymbol{x}_i|\boldsymbol{z}_i)\right]}_{\delta_{\boldsymbol{\beta}}(\boldsymbol{x}_i)}).
\end{aligned}
\tag{9}
$$

Intuitively, based on the discussion in Section 2.1 and Section 3.1, we can approximate the distributions in Eq. (9) by drawing samples from $[\boldsymbol{z}_i|\boldsymbol{x}_i] \sim \mathcal{K}_{T,\boldsymbol{z}_i|\boldsymbol{x}_i} q_{\boldsymbol{\phi}_k}(\boldsymbol{z}_i|\boldsymbol{x}_i)$, $\boldsymbol{z}_i \sim \mathcal{K}_{T,\boldsymbol{z}_i} q_{\boldsymbol{\phi}_k}(\boldsymbol{z}_i)$, to estimate the expectations and hence the learning gradient. Here we learn the DAMC samplers $q_{\boldsymbol{\phi}_k}(\boldsymbol{z}_i|\boldsymbol{x}_i)$ and $q_{\boldsymbol{\phi}_k}(\boldsymbol{z}_i)$ for the posterior and prior sampling chain, respectively. $\boldsymbol{\phi}_k$ represents the current sampler as in Section 3.1; $\mathcal{K}_{T,\boldsymbol{z}_i|\boldsymbol{x}_i}$ and $\mathcal{K}_{T,\boldsymbol{z}_i}$ are the transition kernels for posterior and prior sampling chain. Equivalently, this means to better estimate the learning gradients we can i) first draw approximate posterior and prior MCMC samples from the current $q_{\boldsymbol{\phi}_k}$ model, and ii) update the approximation of the prior $p_{\boldsymbol{\alpha}}(\boldsymbol{z})$ and posterior $p_{\boldsymbol{\theta}}(\boldsymbol{z}|\boldsymbol{x})$ distributions with additional $T$-step LD initialized with $q_{\boldsymbol{\phi}_k}$ samples. These updated samples are closer to $p_{\boldsymbol{\theta}}(\boldsymbol{z}_i|\boldsymbol{x}_i)$ and $p_{\boldsymbol{\alpha}}(\boldsymbol{z}_i)$ compared with short-run LD samples based on our discussion in Section 2.2. Consequently, the diffusion-amortized LD samples provide a generally better estimation of the learning gradients and lead to better performance, as we will show empirically in Section 4. After updating $\boldsymbol{\theta} = (\boldsymbol{\alpha}, \boldsymbol{\beta})$ based on these approximate samples with Eq. (9), we can update $q_{\boldsymbol{\phi}_k}$ with Eq. (8) to distill the sampling chain into $q_{\boldsymbol{\phi}_{k+1}}$. As shown in Fig. 1, we can see that the whole learning procedure iterates between the approximate MLE of $p_{\boldsymbol{\theta}}$ and the amortization of MCMC with $q_{\boldsymbol{\phi}}$. We refer to Appendix A.3 for an extended discussion of this procedure.

After learning the models, we can use either DAMC or LEBM for prior sampling. For DAMC, we may draw samples from $q_{\boldsymbol{\phi}}(\boldsymbol{z}_i)$ with Eq. (5). Prior sampling with LEBM still requires short-run LD initialized from $\mathcal{N}(0, \mathbf{I}_d)$. For posterior sampling, we may sample from $\mathcal{K}_{T,\boldsymbol{z}_i|\boldsymbol{x}_i} q_{\boldsymbol{\phi}_k}(\boldsymbol{z}_i|\boldsymbol{x}_i)$, *i.e.*, first draw samples from DAMC and then run few steps of LD to obtain posterior samples.

---

**Algorithm 1: Learning algorithm of DAMC.**

**Input:** initial parameters $(\boldsymbol{\alpha}, \boldsymbol{\beta}, \boldsymbol{\phi})$; learning rate $\eta = (\eta_{\boldsymbol{\alpha}}, \eta_{\boldsymbol{\beta}}, \eta_{\boldsymbol{\phi}})$; observed examples $\{\boldsymbol{x}^{(i)}\}_{i=1}^{N}$; prob. of uncond. training $p_{\text{uncond}}$ for the DAMC sampler.

**Output:** $\left(\boldsymbol{\theta}^{(K)} = \{\boldsymbol{\alpha}^{(K)}, \boldsymbol{\beta}^{(K)}\}, \boldsymbol{\phi}^{(K)}\right)$.

1 **for** $k = 0 : K - 1$ **do**
2      Sample a minibatch of data $\{\boldsymbol{x}^{(i)}\}_{i=1}^{B}$;
3      **Draw DAMC samples:** For each $\boldsymbol{x}^{(i)}$, draw $\boldsymbol{z}_+^{(i)}$ and $\boldsymbol{z}_-^{(i)}$ from $q_{\boldsymbol{\phi}_k}(\boldsymbol{z}_i|\boldsymbol{x}_i)$.
4      **Prior LD update:** For each $\boldsymbol{x}^{(i)}$, update $\boldsymbol{z}_-^{(i)}$ using Eq. (3), with $\pi(\boldsymbol{z}_i) = p_{\boldsymbol{\alpha}^{(k)}}(\boldsymbol{z}_i)$;
5      **Posterior LD update:** For each $\boldsymbol{x}^{(i)}$, update $\boldsymbol{z}_+^{(i)}$ using Eq. (3), with $\pi(\boldsymbol{z}_i) = p_{\boldsymbol{\beta}^{(k)}}(\boldsymbol{z}_i|\boldsymbol{x}_i)$;
6      **Update $\boldsymbol{\theta}^{(k)}$:** Update $\boldsymbol{\alpha}^{(k)}$ and $\boldsymbol{\beta}^{(k)}$ using Monte-Carlo estimates (*i.e.*, Monte-Carlo average) of Eq. (9) with $\{\boldsymbol{z}_+^{(i)}\}_{i=1}^{B}$ and $\{\boldsymbol{z}_-^{(i)}\}_{i=1}^{B}$.
7      **Update $\boldsymbol{\phi}^{(k)}$:** Update $\boldsymbol{\phi}^{(k)}$ using Eq. (8) with $p_{\text{uncond}}$ and $\{\boldsymbol{z}_+^{(i)}\}_{i=1}^{B}$ as the target.

---

### 3.3 Implementation

In order to efficiently model both $q_{\boldsymbol{\phi}_k}(\boldsymbol{z}_i|\boldsymbol{x}_i)$ and $q_{\boldsymbol{\phi}_k}(\boldsymbol{z}_i)$, we follow the method of [40] to train a single network to parameterize both models, where $q_{\boldsymbol{\phi}_k}(\boldsymbol{z}_i|\boldsymbol{x}_i)$ can be viewed as a conditional DDPM with the embedding of $\boldsymbol{x}_i$ produced by an encoder network as its condition, and $q_{\boldsymbol{\phi}_k}(\boldsymbol{z}_i)$ an unconditional one. For $q_{\boldsymbol{\phi}_k}(\boldsymbol{z}_i)$, we can input a null token $\varnothing$ as its condition when predicting the noise $\boldsymbol{\epsilon}$. We jointly train both models by randomly nullifying the inputs with the probability $p_{\text{uncond}} = 0.2$. During training, we use samples from $q_{\boldsymbol{\phi}_k}(\boldsymbol{z}_i|\boldsymbol{x}_i)$ to initialize both prior and posterior updates for training stability. For the posterior and prior DAMC samplers, we set the number of diffusion steps to 100. The number of iterations in Eq. (8) is set to $M = 6$ throughout the experiments. The LD runs 30 and 60 iterations for posterior and prior updates during training with a step

Table 1: **MSE(↓) and FID(↓) obtained from models trained on different datasets**. The FID scores are computed based on 50k generated images and training images for the first three datasets and 5k images for the CelebA-HQ dataset. The MSEs are computed based on unseen testing images. We highlight our model results in gray color. The best and second-best performances are marked in bold numbers and underlines, respectively; tables henceforth follow this format. *[41] uses a prior model with 4x parameters compared with [22] and ours.

| Model | SVHN | | CelebA | | CIFAR-10 | | CelebA-HQ | |
|---|---|---|---|---|---|---|---|---|
| | MSE | FID | MSE | FID | MSE | FID | MSE | FID |
| VAE [1] | 0.019 | 46.78 | 0.021 | 65.75 | 0.057 | 106.37 | 0.031 | 180.49 |
| 2s-VAE [48] | 0.019 | 42.81 | 0.021 | 44.40 | 0.056 | 72.90 | - | - |
| RAE [49] | 0.014 | 40.02 | 0.018 | 40.95 | 0.027 | 74.16 | - | - |
| NCP-VAE [50] | 0.020 | 33.23 | 0.021 | 42.07 | 0.054 | 78.06 | - | - |
| Adaptive CE* [41] | 0.004 | 26.19 | 0.009 | 35.38 | **0.008** | 65.01 | - | - |
| ABP [51] | - | 49.71 | - | 51.50 | 0.018 | 90.30 | 0.025 | 160.21 |
| SRI [24] | 0.018 | 44.86 | 0.020 | 61.03 | - | - | - | - |
| SRI (L=5) [24] | 0.011 | 35.32 | 0.015 | 47.95 | - | - | - | - |
| LEBM [22] | 0.008 | 29.44 | 0.013 | 37.87 | 0.020 | 70.15 | 0.025 | 133.07 |
| Ours-LEBM | **0.002** | 21.17 | **0.005** | 35.67 | 0.015 | 60.89 | 0.023 | 89.54 |
| Ours-DAMC | | **18.76** | | **30.83** | | **57.72** | | **85.88** |

size of $s = 0.1$. For test time sampling from $\mathcal{K}_{T,\boldsymbol{z}_i|\boldsymbol{x}_i} q_{\boldsymbol{\phi}_k}(\boldsymbol{z}_i|\boldsymbol{x}_i)$, $T = 10$ for the additional LD. For a fair comparison, we use the same LEBM and generator as in [22, 41] for all the experiments. We summarize the learning algorithm in Algorithm 1. Please see Appendices B and C for network architecture and further training details, as well as the pytorch-style pseudocode of the algorithm.

# 4    Experiments

In this section, we are interested in the following questions: (i) How does the proposed method compare with its previous counterparts (*e.g.*, purely MCMC-based or variational methods)? (ii) How is the scalability of this method? (iii) How are the time and parameter efficiencies? (iv) Does the proposed method provide a desirable latent space? To answer these questions, we present a series of experiments on benchmark datasets including MNIST [42], SVHN [43], CelebA64 [44], CIFAR-10 [45], CelebAMask-HQ [46], FFHQ [10] and LSUN-Tower [47]. As to be shown, the proposed method demonstrates consistently better performance in various experimental settings compared with previous methods. We refer to Appendix D for details about the experiments.

## 4.1    Generation and Inference: Prior and Posterior Sampling

**Generation and reconstruction**    We evaluate the quality of the generated and reconstructed images to examine the sampling quality of DAMC. Specifically, we would like to check i) how well does DAMC fit the seen data, ii) does DAMC provide better MCMC samples for learning LEBM and iii) the generalizability of DAMC on unseen data. We check the goodness of fit of DAMC by evaluating the quality of the images generated with DAMC prior samples. If DAMC does provide better MCMC samples for learning LEBM, we would expect better fitting of data and hence an improved generation quality of LEBM. We evaluate the performance of posterior sampling given unseen testing images by examining the reconstruction error on testing data. We benchmark our model against a variety of previous methods in two groups. The first group covers competing methods that adopt the variational learning scheme, including VAE [1], as well as recent two-stage methods such as 2-stage VAE [48], RAE [49] and NCP-VAE [50], whose prior distributions are learned with posterior samples in a second stage after the generator is trained. The second group includes methods that adopt MCMC-based sampling. It includes Alternating Back-Propagation (ABP) [51], Short-Run Inference (SRI) from [24] and the vanilla learning method of LEBM [22], which relies on short-run LD for both posterior and prior sampling. We also compare our method with the recently proposed Adaptive CE [41]. It learns a series of LEBMs adaptively during training, while these LEBMs are sequentially updated by density ratio estimation instead of MLE. To make fair comparisons, we follow the same evaluation protocol as in [22, 41].

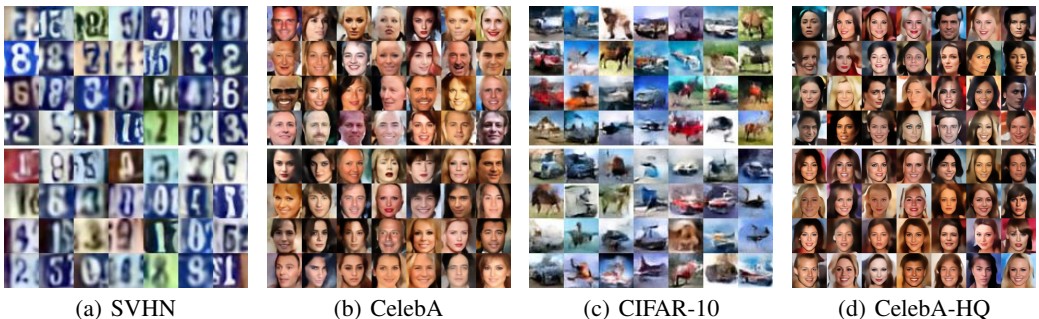

| (a) SVHN | (b) CelebA | (c) CIFAR-10 | (d) CelebA-HQ |

Figure 2: **Samples generated from the DAMC sampler and LEBM** trained on SVHN, CelebA, CIFAR-10 and CelebA-HQ datasets. In each sub-figure, the first four rows are generated by the DAMC sampler. The last four rows are generated by LEBM trained with the DAMC sampler.

For generation, we report the FID scores [52] in Table 1. We observe that i) the DAMC sampler, denoted as Ours-DAMC, provides superior generation performance compared to baseline models, and ii) the LEBM learned with samples from DAMC, denoted as Ours-LEBM, demonstrates significant performance improvement compared with the LEBM trained with short-run LD, denoted as LEBM. These results confirm that DAMC is a reliable sampler and indeed partly addresses the learning issue of LEBM caused by short-run LD. We would like to point out that the improvement is clearer on the CelebAMask-HQ dataset, where the input data is of higher dimension ($256 \times 256$) and contains richer details compared with other datasets. This illustrates the superiority of DAMC sampler over short-run LD when the target distribution is potentially highly multi-modal. We show qualitative results of generated samples in Fig. 2, where we observe that our method can generate diverse, sharp and high-quality samples. For reconstruction, we compare our method with baseline methods in terms of MSE in Table 1. We observe that our method demonstrates competitive reconstruction error, if not better, than competing methods do. Additional qualitative results of generation and reconstruction are presented in Appendices E.2 and E.3.

**GAN inversion** We have examined the scalability of our method on the CelebAMask-HQ dataset. Next, we provide more results on high-dimensional and highly multi-modal data by performing GAN inversion [54] using the proposed method. Indeed, we may regard GAN inversion as an inference problem and a special case of posterior sampling. As a suitable testbed, the StyleGAN structure [10] is specifically considered as our generator in the experiments: [53] points out that to effectively infer the latent representation of a given image, the GAN inversion method needs to consider an extended latent space of StyleGAN, consisting of

| Model | FFHQ | | LSUN-T | |
|---|---|---|---|---|
| | MSE | FID | MSE | FID |
| Opt. [53] | 0.055 | 149.39 | 0.080 | 240.11 |
| Enc. [16] | 0.028 | 62.32 | 0.079 | 132.41 |
| [22] w/ 1x | 0.054 | 149.21 | 0.072 | 239.51 |
| [22] w/ 2x | 0.039 | 101.59 | 0.066 | 163.20 |
| [22] w/ 4x | 0.032 | 84.64 | 0.059 | 111.53 |
| Ours | **0.025** | **52.85** | **0.059** | **80.42** |

Table 2: **MSE(↓) and FID(↓) for GAN inversion on different datasets**. Opt. and Enc. denotes the optimization-based and encoder-based methods.

14 different 512-dimensional latent vectors. We attempt to use the DAMC sampler for GAN inversion. We benchmark our method against i) learning an encoder that maps a given image to the latent space [16], which relates to the variational methods for posterior inference, ii) optimizing a random initial latent code by minimizing the reconstruction error and perceptual loss [53], which can be viewed as a variant of LD sampling, and iii) optimizing the latent code by minimizing both the objectives used in ii) and the energy score provided by LEBM. We use the pretrained weights provided by [18] for the experiments. Both the DAMC sampler and the encoder-based method are augmented with 100 post-processing optimization iterations. We refer to Appendix D for more experiment details. We test LEBM-based inversion with different optimization iterations. To be specific, 1x, 2x, and 4x represent 100, 200, and 400 iterations respectively. We can see in Table 2 that DAMC performs better than all the baseline methods on the unseen testing data, which supports the efficacy of our method in high-dimensional settings. We provide qualitative results in Fig. 3.

**Parameter efficiency and sampling time** One potential disadvantage of our method is its parameter inefficiency for introducing an extra DDPM. Fortunately, our models are in the latent space

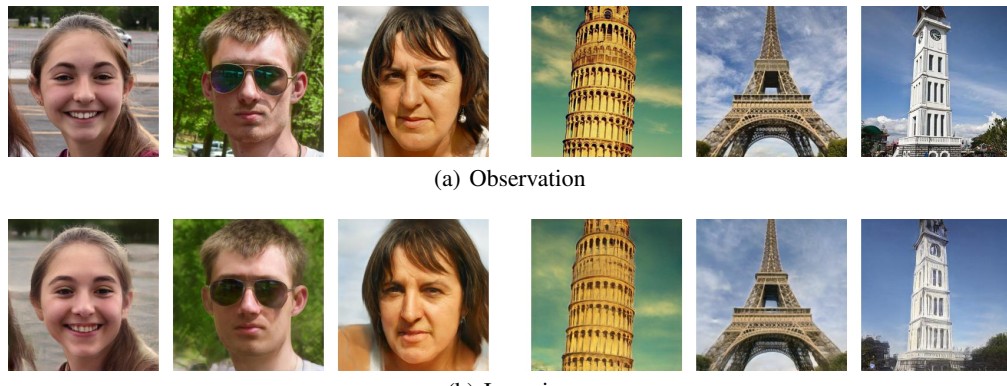

(a) Observation

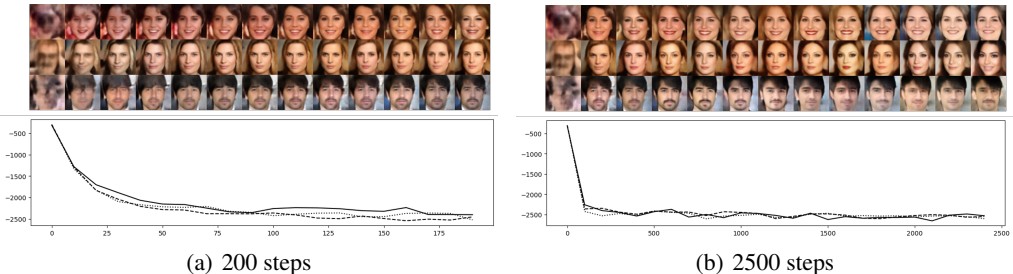

(b) Inversion

Figure 3: **Qualitative results of StyleGAN inversion using the DAMC sampler.** In each sub-figure, the left panel contain samples from the FFHQ dataset, and the right panel contains samples from the LSUN-T dataset.

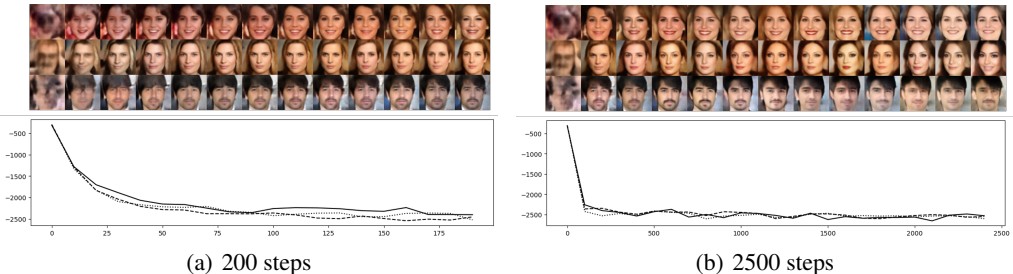

(a) 200 steps  (b) 2500 steps

Figure 4: **Transition of Markov chains initialized from $\mathcal{N}(0, \mathbf{I}_d)$ towards $p_{\boldsymbol{\alpha}}(\boldsymbol{z})$.** We present results by running LD for 200 and 2500 steps. In each sub-figure, the top panel displays the trajectory in the data space uniformly sampled along the chain. The bottom panel shows the energy score $f_{\boldsymbol{\alpha}}(\boldsymbol{z})$ over the iterations.

so the network is lightweight. To be specific, on CIFAR-10 dataset the number of parameters in the DDPM is only around $10\%$ (excluding the encoder) of those in the generator. The method has competitive time efficiency. With the batch size of $64$, the DAMC prior sampling takes $0.3s$, while $100$ steps of short-run LD with LEBM takes $0.2s$. The DAMC posterior sampling takes $1.0s$, while LEBM takes $8.0s$. Further discussions about the limitations can be found in the Appendix G.1.

## 4.2 Analysis of Latent Space

**Long-run langevin transition** In this section, we examine the energy landscape induced by the learned LEBM. We expect that a well-trained $p_{\boldsymbol{\alpha}}(\boldsymbol{z})$ fueled by better prior and posterior samples from the DAMC sampler would lead to energy landscape with regular geometry. In Fig. 4, we visualize the transition of LD initialized from $\mathcal{N}(0, \mathbf{I}_d)$ towards $p_{\boldsymbol{\alpha}}(\boldsymbol{z})$ on the model trained on the CelebA dataset. Additional visualization of transitions on SVHN and CIFAR-10 datasets can be found in the Appendix E.4. The LD iterates for 200 and 2500 steps, which is longer than the LD within each training iteration (60 steps). For the 200-step set-up, we can see that the generation quality quickly improves by exploring the local modes (demonstrating different facial features, *e.g.*, hairstyle, facial expression and lighting). For the 2500-step long-run set-up, we can see that the LD produces consistently valid results without the oversaturating issue of the long-run chain samples [23]. These observations provide empirical evidence that the LEBM is well-trained.

**Anomaly detection** We further evaluate how the LEBM learned by our method could benefit the anomaly detection task. With properly learned models, the posterior $p_{\boldsymbol{\theta},\boldsymbol{\phi}}(\boldsymbol{z}|\boldsymbol{x})$ could form a discriminative latent space that has separated probability densities for in-distribution (normal) and out-of-distribution (anomalous) data. Given the testing sample $\boldsymbol{x}$, we use un-normalized log joint density $p_{\boldsymbol{\theta},\boldsymbol{\phi}}(\boldsymbol{z}|\boldsymbol{x}) \propto p_{\boldsymbol{\theta},\boldsymbol{\phi}}(\boldsymbol{x}, \boldsymbol{z}) \approx p_{\boldsymbol{\beta}}(\boldsymbol{x}|\boldsymbol{z})p_{\boldsymbol{\alpha}}(\boldsymbol{z})|_{\boldsymbol{z} \sim \mathcal{K}_{T,\boldsymbol{z}|\boldsymbol{x}} q_{\boldsymbol{\phi}}(\boldsymbol{z}|\boldsymbol{x})}$ as our decision function. This means that we draw samples from $\mathcal{K}_{T,\boldsymbol{z}|\boldsymbol{x}} q_{\boldsymbol{\phi}}(\boldsymbol{z}|\boldsymbol{x})$ and compare the corresponding reconstruction errors and energy scores. A higher value of log joint density indicates a higher probability of the test sample being a normal sample. To make fair comparisons, we follow the experimental settings in [22, 41, 55, 56] and train our model on MNIST with one class held out as an anomalous class. We consider the

Table 3: **AUPRC(↑) scores for unsupervised anomaly detection on MNIST**. Numbers are taken from [41]. Results of our model are averaged over the last 10 trials to account for variance.

| Heldout Digit | 1 | 4 | 5 | 7 | 9 |
|---|---|---|---|---|---|
| VAE [1] | 0.063 | 0.337 | 0.325 | 0.148 | 0.104 |
| ABP [51] | $0.095 \pm 0.03$ | $0.138 \pm 0.04$ | $0.147 \pm 0.03$ | $0.138 \pm 0.02$ | $0.102 \pm 0.03$ |
| MEG [55] | $0.281 \pm 0.04$ | $0.401 \pm 0.06$ | $0.402 \pm 0.06$ | $0.290 \pm 0.04$ | $0.342 \pm 0.03$ |
| BiGAN-$\sigma$ [56] | $0.287 \pm 0.02$ | $0.443 \pm 0.03$ | $0.514 \pm 0.03$ | $0.347 \pm 0.02$ | $0.307 \pm 0.03$ |
| LEBM [22] | $0.336 \pm 0.01$ | $0.630 \pm 0.02$ | $0.619 \pm 0.01$ | $0.463 \pm 0.01$ | $0.413 \pm 0.01$ |
| Adaptive CE [41] | $0.531 \pm 0.02$ | $0.729 \pm 0.02$ | $0.742 \pm 0.01$ | $0.620 \pm 0.02$ | $0.499 \pm 0.01$ |
| Ours | $\mathbf{0.684 \pm 0.02}$ | $\mathbf{0.911 \pm 0.01}$ | $\mathbf{0.939 \pm 0.02}$ | $\mathbf{0.801 \pm 0.01}$ | $\mathbf{0.705 \pm 0.01}$ |

Table 4: **Ablation study on CIFAR-10 dataset**. VI denotes learning LEBM using variational methods. SR denotes learning LEBM with short-run LD. DAMC-G replaces the LEBM in DAMC-LEBM with a standard Gaussian distribution. NALR denotes the non-amortized DDPM setting. For each set-up, we provide results using the vanilla sampling method, denoted as V, and the ones using the DAMC sampler, denoted as D.

| Model | VI-LEBM | | SR-LEBM | | DAMC-G | | NALR-LEBM | | DAMC-LEBM | |
|---|---|---|---|---|---|---|---|---|---|---|
| | V. | D. | V. | D. | V. | D. | V. | D. | V. | D. |
| MSE | 0.054 | - | 0.020 | - | 0.018 | 0.015 | 0.028 | 0.016 | 0.021 | **0.015** |
| FID | 78.06 | - | 70.15 | - | 90.30 | 66.93 | 68.52 | 64.38 | 60.89 | **57.72** |

baseline models that employ MCMC-based or variational inferential mechanisms. Table 3 shows the results of AUPRC scores averaged over the last 10 trials. We observe significant improvements in our method over the previous counterparts.

## 4.3 Ablation Study

In this section, we conduct ablation study on several variants of the proposed method. Specifically, we would like to know: i) what is the difference between the proposed method and directly training a DDPM in a fixed latent space? ii) What is the role of LEBM in this learning scheme? iii) Does DAMC effectively amortize the sampling chain? We use CIFAR-10 dataset for the ablative experiments to empirically answer these questions. More ablation studies can be found in Appendix F.

**Non-Amortized DDPM vs. DAMC**   We term directly training a DDPM in a fixed latent space as the non-amortized DDPM. To analyze the difference between non-amortized DDPM and DAMC, we first train a LEBM model with persistent long-run chain sampling [57] and use the trained model to obtain persistent samples for learning the non-amortized DDPM. In short, the non-amortized DDPM can be viewed as directly distilling the long-run MCMC sampling process, instead of progressively amortizing the chain. We present the FID and MSE of the non-amortized model (NALR) in Table 4. We observe that directly amortizing the long-run chain leads to degraded performance compared with the proposed method. The results are consistently worse for both posterior and prior sampling and the learned LEBMs, which verify the effectiveness of the proposed iterative learning scheme.

**The contribution of LEBM**   One may argue that since we have the DAMC as a powerful sampler, it might not be necessary to jointly learn LEBM in the latent space. To demonstrate the necessity of this joint learning scheme, we train a variant of DAMC by replacing the LEBM with a Gaussian prior. The results are presented in Table 4. We observe that models trained with non-informative Gaussian prior obtain significantly worse generation results. It suggests that LEBM involved in the learning iteration provides positive feedback to the DAMC sampler. Therefore, we believe that it is crucial to jointly learn the DAMC and LEBM.

**Vanilla sampling vs. DAMC**   We compare the vanilla sampling process of each model with DAMC. The vanilla sampling typically refers to short-run or long-run LD initialized with $\mathcal{N}(0, \mathbf{I}_d)$. We also provide results of learning LEBM using variational methods for comparison. We can see in Table 4 that sampling with DAMC shows significantly better scores of the listed models, compared with vanilla sampling. The result is even better than that of the persistent chain sampler (V. of NALR-LEBM). This indicates that DAMC effectively amortizes the sampling chain. Comparing DAMC sampler with the variational sampler also indicates that DAMC is different from general variational approximation: it benefits from its connection with LD and shows better expressive power.

# 5 Related Work

**Energy-based prior model**   EBMs [23, 24, 58–60] play an important role in generative modeling. Pang et al. [22] propose to learn an EBM as a prior model in the latent space of DLVMs; it greatly improves the model expressivity over those with non-informative priors and brings strong performance on downstream tasks, *e.g.*, image segmentation, text modeling, molecule generation, and trajectory prediction [12, 15, 61, 62]. However, learning EBMs or latent space EBMs requires MCMC sampling to estimate the learning gradients, which needs numerous iterations to converge when the target distributions are high-dimensional or highly multi-modal. Typical choices of sampling with non-convergent short-run MCMC [24] in practice can lead to poor generation quality, malformed energy landscapes [15, 24, 25], biased estimation of the model parameter and instability in training [23, 25, 59, 60]. In this work, we consider learning valid amortization of the long-run MCMC for energy-based priors; the proposed model shows reliable sampling quality in practice.

**Denoising diffusion probabilistic model**   DDPMs [7, 8, 31], originating from [5], learn the generative process by recovering the observed data from a sequence of noise-perturbed versions of the data. The learning objective can be viewed as a variant of the denoising score matching objective [35]. As pointed out in [5, 8], the sampling procedure of DDPM with $\epsilon$-prediction parametrization resembles LD of an EBM; $\epsilon$ (predicted noise) plays a similar role to the gradient of the log density [8]. To be specific, learning a DDPM with $\epsilon$-prediction parameterization is equivalent to fitting the finite-time marginal of a sampling chain resembling annealed Langevin dynamics [7, 8, 31]. Inspired by this connection, we propose to amortize the long-run MCMC in learning energy-based prior by iteratively distilling the short-run chain segments with a DDPM-based sampler. We show empirically and theoretically that the learned sampler is valid for long-run chain sampling.

**Amortized MCMC**   The amortized MCMC technique is formally brought up by Li et al. [36], which incorporates feedback from MCMC back to the parameters of the amortizer distribution $q_\phi$. It is concurrently and independently proposed by Xie et al. [37] as the MCMC teaching framework. Methods under this umbrella term [36–38, 63–66] generally learns the amortizer by minimizing the divergence (typically the KLD) between the improved distribution and its initialization, i.e., $\mathcal{D}[\mathcal{K}_T q_{\phi_{k-1}} || q_\phi]$, where $\mathcal{K}_T$ represents $T$-step MCMC transition kernel and $q_{\phi_{k-1}}$ represents the current amortizer. The diffusion-based amortization proposed in this work can be viewed as an instantiation of this framework, while our focus is on learning the energy-based prior. Compared with previous methods, our method i) specifically exploits the connection between EBMs and DDPMs and is suitable for amortizing the prior and posterior sampling MCMC of energy-based prior, and ii) resides in the lower-dimensional latent space and enables faster sampling and better convergence.

**More methods for learning EBM**   Several techniques other than short-run MCMC have been proposed to learn the EBM. In the seminal work, Hinton [67] proposes to initialize Markov chains using real data and run several steps of MCMC to obtain samples from the model distribution. Tieleman [57] proposes to start Markov chains from past samples in the previous sampling iteration, known as Persistent Contrastive Divergence (PCD) or persistent chain sampling, to mimic the long-run sampling chain. Nijkamp et al. [23] provide comprehensive discussions about tuning choices for LD such as the step size $s$ and sampling steps $T$ to obtain stable long-run samples for persistent training. [59, 60] employ a hybrid of persistent chain sampling and short-run sampling by maintaining a buffer of previous samples. The methods draw from the buffer or initialize the short-run chain with noise distribution with some pre-specified probability. Another branch of work, stemmed from [68], considers discriminative contrastive estimation to avoid MCMC sampling. Gao et al. [69] use a normalizing flow [4] as the base distribution for contrastive estimation. Aneja et al. [50] propose to estimate the energy-based prior model based on the prior of a pre-trained VAE [70] by noise contrastive estimation. More recently, Xiao and Han [41] learn a sequence of EBMs in the latent space with adaptive multi-stage NCE to further improve the expressive power of the model.

# 6 Conclusion

In this paper, we propose the DAMC sampler and develop a novel learning algorithm for LEBM based on it. We provide theoretical and empirical evidence for the effectiveness of our method. We notice that our method can be applied to amortizing MCMC sampling of unnormalized continuous densities in general. It can also be applied to sampling posterior distributions of continuous latent variables in general latent variable models. We would like to explore these directions in future work.

## Acknowledgements

Y. N. Wu was supported by NSF DMS-2015577. We would like to thank the anonymous reviewers for their constructive comments.

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

# A  Theoretical Discussion

## A.1  Monotonically Decreasing KLD

We state in the main text that $\mathcal{D}[q_{\phi_k}||\pi] \leqslant \mathcal{D}[q_{\phi_{k-1}}||\pi]$, where $\pi$ is the stationary distribution. To show this, we first provide a proof of $\mathcal{D}[\pi_{t+T}||\pi] \leqslant \mathcal{D}[\pi_t||\pi]$, where $\pi_t$ and $\pi_{t+T}$ are the distributions of $z$ at $t$-th and $(t+T)$-th iteration, respectively. This is a known result from [39], and we include it here for completeness.

$$
\begin{aligned}
\mathcal{D}[\pi_t||\pi] &= \mathbb{E}_{\pi_t(z_t)}\left[\log\frac{\pi_t(z_t)}{\pi(z_t)}\right] = \mathbb{E}_{\pi_t(z_t),\mathcal{K}(z_{t+1}|z_t)}\left[\log\frac{\pi_t(z_t)\mathcal{K}(z_{t+1}|z_t)}{\pi(z_t)\mathcal{K}(z_{t+1}|z_t)}\right] \\
&\overset{(i)}{=} \mathbb{E}_{\pi_{t+1}(z_{t+1}),\mathcal{K'}_{\pi_{t+1}}(z_t|z_{t+1})}\left[\log\frac{\mathcal{K'}_{\pi_{t+1}}(z_t|z_{t+1})\pi_{t+1}(z_{t+1})}{\mathcal{K'}_{\pi}(z_t|z_{t+1})\pi(z_{t+1})}\right] \\
&= \mathcal{D}[\pi_{t+1}||\pi] + \mathbb{E}_{\pi_{t+1}(z_{t+1})}\mathcal{D}[\mathcal{K'}_{\pi_{t+1}}(z_t|z_{t+1})||\mathcal{K'}_{\pi}(z_t|z_{t+1})] \\
&\overset{(ii)}{\geqslant} \mathcal{D}[\pi_{t+1}||\pi],
\end{aligned}
\tag{10}
$$

where we denote the forward-time transition kernel as $\mathcal{K}$ and the reverse-time kernel as $\mathcal{K'}$. (i) holds because we are just re-factorizing the joint density of $[z_t, z_{t+1}]$: $\pi_t(z_t)\mathcal{K}(z_{t+1}|z_t) = \mathcal{K'}_{\pi_{t+1}}(z_t|z_{t+1})\pi_{t+1}(z_{t+1})$ and $\pi(z_t)\mathcal{K}(z_{t+1}|z_t) = \mathcal{K'}_{\pi}(z_t|z_{t+1})\pi(z_{t+1})$. (ii) holds because the KLD is non-negative. We can see that $\mathcal{D}[\pi_{t+T}||\pi] \leqslant \mathcal{D}[\pi_t||\pi]$ is a direct result from Eq. (10), and that $\pi_t \to \pi$ as $t \to \infty$ under proper conditions [32].

In the main text, we describe the update rule of the sampler $q_\phi$ as follows:

$$
q_{\phi_k} \leftarrow \underset{q_\phi \in \mathcal{Q}}{\arg\min}\, \mathcal{D}[q_{\phi_{k-1},T}||q_\phi], \; q_{\phi_{k-1},T} := \mathcal{K}_T q_{\phi_{k-1}}, \; q_{\phi_0} \approx \pi_0, \; k = 0, ..., K-1.
\tag{11}
$$

In the ideal case, we can assume that the objective in Eq. (6) is properly optimized and that $\{q_\phi\}$ is expressive enough to parameterize each $q_{\phi_{k-1},T}$. With $q_{\phi_0} \approx \pi_0$ and $q_{\phi_k} \approx \mathcal{K}_T q_{\phi_{k-1}}$, we can conclude that $\mathcal{D}[q_{\phi_k}||\pi] \leqslant \mathcal{D}[q_{\phi_{k-1}}||\pi]$ for each $k = 1, ..., K$ according to Eq. (10). We will discuss in the following section the scenario where we apply gradient-based methods to minimize $\mathcal{D}[q_{\phi_{k-1},T}||q_\phi]$ and approximate Eq. (6) for the update from $q_{\phi_{k-1}}$ to $q_{\phi_k}$.

## A.2  Discussion about Diffusion-Based Amortization

We can see that the statement in Appendix A.1 holds when $q_{\phi_k}$ is a close approximation of $q_T := \mathcal{K}_T q_{\phi_{k-1}}$. This motivates our choice of employing DDPM to amortize the LD transition, considering its capability of close approximation to the given distribution $\{q_T\}$. Based on the derivation of DDPM learning objective in [1], we know that

$$
\underset{q_\phi}{\arg\min}\, \mathcal{D}[q_T||q_\phi] = \underset{q_\phi}{\arg\min}\, -\mathcal{H}(q_T) + \mathcal{H}(q_T, q_\phi) = \underset{q_\phi}{\arg\min}\, -\mathbb{E}_{q_T}\left[\log q_\phi\right]
$$

$$
\leqslant \underset{q_\phi}{\arg\min}\, \mathbb{E}_{\epsilon,\lambda}\left[\|\epsilon_\phi(z_\lambda) - \epsilon\|_2^2\right] \approx \underset{q_\phi}{\arg\min}\, \frac{1}{N}\sum_{j=1}^{N}\left[\|\epsilon_\phi(z_{j,\lambda_j}) - \epsilon_j\|_2^2\right],
\tag{12}
$$

where $z_\lambda$ is draw from $q(z_\lambda|z_0) = \mathcal{N}(z_\lambda; \beta_\lambda z_0, \sigma_\lambda^2 \mathbf{I}_d)$. $z_0 \sim q_T$. $\epsilon \sim \mathcal{N}(\mathbf{0}, \mathbf{I}_d)$ and $\lambda$ is drawn from a distribution of log noise-to-signal ratio $p(\lambda)$. In practice, we use Monte-Carlo average to approximate the objective and employ a gradient-based update rule for $q_\phi$:

$$
\phi_{k-1}^{(i+1)} \leftarrow \phi_{k-1}^{(i)} - \eta\nabla_\phi \frac{1}{N}\sum_{j=1}^{N}\left[\|\epsilon_\phi(z_{j,\lambda_j})\epsilon_j\|_2^2\right], \; \phi_k^{(0)} \leftarrow \phi_{k-1}^{(M)}, \; i = 0, 1, ..., M-1.
\tag{13}
$$

This can be viewed as a M-estimation of $\phi$. Recall that $q_T = \mathcal{K}_T q_{\phi_{k-1}}$, we can construct $\phi_k = \left(\phi_{k-1}, \tilde{\phi}\right)$ to minimize the KLD, where $\tilde{\phi}$ models the transition kernel $\mathcal{K}_T$. Therefore, initializing $q_\phi$ to be optimized with $q_{\phi_{k-1}}$, we are effectively maximizing $\mathcal{L}(\tilde{\phi}) = \frac{1}{N}\sum_{j=1}^{N}\log p_{\tilde{\phi}}(\hat{z}_j|z_j)$,

where $\{\boldsymbol{z}_j\}$ are from $q_{\boldsymbol{\phi}_{k-1}}$ and $\{\hat{\boldsymbol{z}}_j\}$ are from $\mathcal{K}_T q_{\boldsymbol{\phi}_{k-1}}$. Let $\tilde{\boldsymbol{\phi}}$ be the result of M-estimation and $\tilde{\boldsymbol{\phi}}^*$ be the true target parameter. Then based on the derivation in [25], asymptotically we have

$$\sqrt{N}\left(\tilde{\boldsymbol{\phi}} - \tilde{\boldsymbol{\phi}}^*\right) \to \mathcal{N}\left(0, \mathcal{I}\left(\tilde{\boldsymbol{\phi}}^*\right)^{-1}\right), \tag{14}$$

where $\mathcal{I}\left(\tilde{\boldsymbol{\phi}}^*\right) = \mathbb{E}_{\hat{\boldsymbol{z}},\boldsymbol{z}}\left[-\nabla^2 \log p_{\tilde{\boldsymbol{\phi}}}(\hat{\boldsymbol{z}}|\boldsymbol{z})\right]$ is the Fisher information matrix. This interpretation tells us that i) when the sample size $N$ is large, the estimation $\tilde{\boldsymbol{\phi}}$ is asymptotically unbiased, and ii) if we want to obtain the estimation $\tilde{\boldsymbol{\phi}}$ with a few gradient-based updates, then the eigenvalues of the Fisher information matrix would be relatively small but non-zero. ii) suggests that $\mathcal{K}_T q_{\boldsymbol{\phi}_{k-1}}$ should be significantly different from $q_{\boldsymbol{\phi}_{k-1}}$, which is confirmed by [38] and our preliminary experiments, but it should not be too far away because that would require more gradient-based updates. We find that setting $T = 30$ and $M = 6$ works well in the experiments.

## A.3 Further Discussion about the Learning Algorithm

For completeness, we first derive the learning gradients for updating $\boldsymbol{\theta}$.

$$
\begin{aligned}
\nabla_{\boldsymbol{\theta}} \log p_{\boldsymbol{\theta}}(\boldsymbol{x}) &= \frac{1}{p_{\boldsymbol{\theta}}(\boldsymbol{x})} \nabla_{\boldsymbol{\theta}} \int_{\boldsymbol{z}} p_{\boldsymbol{\theta}}(\boldsymbol{x}, \boldsymbol{z}) d\boldsymbol{z} = \int_{\boldsymbol{z}} \frac{p_{\boldsymbol{\theta}}(\boldsymbol{x}, \boldsymbol{z})}{p_{\boldsymbol{\theta}}(\boldsymbol{x})} \nabla_{\boldsymbol{\theta}} \log p_{\boldsymbol{\theta}}(\boldsymbol{x}, \boldsymbol{z}) d\boldsymbol{z} \\
&= \mathbb{E}_{p_{\boldsymbol{\theta}}(\boldsymbol{z}|\boldsymbol{x})} \left[\nabla_{\boldsymbol{\theta}} \log p_{\boldsymbol{\theta}}(\boldsymbol{z}, \boldsymbol{x})\right] \\
&= \left(\mathbb{E}_{p_{\boldsymbol{\theta}}(\boldsymbol{z}|\boldsymbol{x})} \left[\nabla_{\boldsymbol{\alpha}} \log p_{\boldsymbol{\alpha}}(\boldsymbol{z})\right], \mathbb{E}_{p_{\boldsymbol{\theta}}(\boldsymbol{z}|\boldsymbol{x})} \left[\nabla_{\boldsymbol{\beta}} \log p_{\boldsymbol{\beta}}(\boldsymbol{x}|\boldsymbol{z})\right]\right) \\
&= (\underbrace{\mathbb{E}_{p_{\boldsymbol{\theta}}(\boldsymbol{z}|\boldsymbol{x})} \left[\nabla_{\boldsymbol{\alpha}} f_{\boldsymbol{\alpha}}(\boldsymbol{z})\right] - \mathbb{E}_{p_{\boldsymbol{\alpha}}(\boldsymbol{z})} \left[\nabla_{\boldsymbol{\alpha}} f_{\boldsymbol{\alpha}}(\boldsymbol{z})\right]}_{\delta_{\boldsymbol{\alpha}}(\boldsymbol{x})}, \underbrace{\mathbb{E}_{p_{\boldsymbol{\theta}}(\boldsymbol{z}|\boldsymbol{x})} \left[\nabla_{\boldsymbol{\beta}} \log p_{\boldsymbol{\beta}}(\boldsymbol{x}|\boldsymbol{z})\right]}_{\delta_{\boldsymbol{\beta}}(\boldsymbol{x})}).
\end{aligned} \tag{15}
$$

We can see that $\boldsymbol{\theta}$ is estimated in a MLE-by-EM style. The learning gradient is the same as that of directly maximizing the observed data likelihood, while we need to approximate the expectations in Eq. (15). Estimating the expectations is like the E-step, and update $\boldsymbol{\theta}$ with Eq. (15) is like the M-step in the EM algorithm. The proposed diffusion-based amortization brings better estimation of the expectations in the E-step, and incorporate the feedback from the M-step by running prior and posterior sampling LD as follows

$$
\begin{aligned}
\boldsymbol{z}_{t+1} &= \boldsymbol{z}_t + \frac{s^2}{2} \nabla_{\boldsymbol{z}_t} \underbrace{\left(f_{\boldsymbol{\alpha}}(\boldsymbol{z}_t) - \frac{1}{2}\|\boldsymbol{z}_t\|_2^2\right)}_{\log p_{\boldsymbol{\alpha}}(\boldsymbol{z}_t)} + s\boldsymbol{w}_t, \\
\boldsymbol{z}_{t+1} &= \boldsymbol{z}_t + \frac{s^2}{2} \nabla_{\boldsymbol{z}_t} \underbrace{\left(-\frac{\|\boldsymbol{x} - g_{\boldsymbol{\beta}}(\boldsymbol{z}_t)\|_2^2}{2\sigma^2} + f_{\boldsymbol{\alpha}}(\boldsymbol{z}_t) - \frac{1}{2}\|\boldsymbol{z}_t\|_2^2\right)}_{\log p_{\boldsymbol{\theta}}(\boldsymbol{z}|\boldsymbol{x}) = \log p_{\boldsymbol{\theta}}(\boldsymbol{x}, \boldsymbol{z}) + C} + s\boldsymbol{w}_t,
\end{aligned} \tag{16}
$$

to obtain training data. Here $t = 0, 1, ..., T$. $\boldsymbol{z}_0 \sim q_{\boldsymbol{\phi}}(\boldsymbol{z}|\boldsymbol{x})$ for posterior sampling and $\boldsymbol{z}_0 \sim q_{\boldsymbol{\phi}}(\boldsymbol{z})$ for prior sampling. $\boldsymbol{w}_t \sim \mathcal{N}(\mathbf{0}, \mathbf{I}_d)$. Note that we plug-in $p_{\boldsymbol{\theta}}(\boldsymbol{x}, \boldsymbol{z})$ for the target distribution of posterior sampling LD. This is because given the observed data $\boldsymbol{x}$, by Bayes' rule we know that $p_{\boldsymbol{\theta}}(\boldsymbol{z}|\boldsymbol{x}) \propto p_{\boldsymbol{\theta}}(\boldsymbol{x}, \boldsymbol{z}) = p_{\boldsymbol{\beta}}(\boldsymbol{x}|\boldsymbol{z}) p_{\boldsymbol{\alpha}}(\boldsymbol{z})$. The whole learning iteration can be viewed as a variant of the variational EM algorithm [71].

# B  Network Architecture and Training Details

**Architecture**  The energy score network $f_\alpha$ uses a simple fully-connected structure throughout the experiments. We describe the architecture in details in Table S1. The generator network has a simple deconvolution structure similar to DCGAN [72] shown in Table S2. The denoising diffusion model is implemented by a light-weight MLP-based U-Net [73] structure (Table S3). The encoder network to embed the observed images has a fully convolutional structure [74], as shown in Table S4.

Table S1: **Network structures of the energy score network.** LReLU denotes the Leaky ReLU activation function. The slope in Leaky ReLU is set to 0.2. For SVHN and CelebA datasets, we use `nz=100`. For the CIFAR-10 and CelebA-HQ datasets, we use `nz=128`. We use `nz=8` for anomaly detection on the MNIST dataset, and `nz=7168` for GAN inversion. We use `ndf=512` for GAN inversion and `ndf=200` for the rest experiments.

| Layers | Out Size |
|---|---|
| Input: $z$ | nz $\in \{8, 100, 128, 7168\}$ |
| Linear, LReLU | ndf $\in \{200, 512\}$ |
| Linear, LReLU | ndf $\in \{200, 512\}$ |
| Linear | 1 |

Table S2: **Network structures of the generator networks** used for the SVHN, CelebA, CIFAR-10, CelebA-HQ and MNIST (from top to bottom) datasets. For GAN inversion, we use the StyleGAN [10] structure as our generator network. ConvT($n$) indicates a transposed convolutional operation with $n$ output channels. We use `ngf=64` for the SVHN dataset and `ngf=128` for the rest. LReLU indicates the Leaky-ReLU activation function. The slope in Leaky ReLU is set to be 0.2.

| Layers | Out Size | Stride |
|---|---|---|
| Input: $z$ | 1x1x100 | - |
| 4x4 ConvT(ngf x 8), LReLU | 4x4x(ngf x 8) | 1 |
| 4x4 ConvT(ngf x 4), LReLU | 8x8x(ngf x 4) | 2 |
| 4x4 ConvT(ngf x 2), LReLU | 16x16x(ngf x 2) | 2 |
| 4x4 ConvT(3), Tanh | 32x32x3 | 2 |
| **Layers** | **Out Size** | **Stride** |
| Input: $z$ | 1x1x100 | - |
| 4x4 ConvT(ngf x 8), LReLU | 4x4x(ngf x 8) | 1 |
| 4x4 ConvT(ngf x 4), LReLU | 8x8x(ngf x 4) | 2 |
| 4x4 ConvT(ngf x 2), LReLU | 16x16x(ngf x 2) | 2 |
| 4x4 ConvT(ngf x 1), LReLU | 32x32x(ngf x 1) | 2 |
| 4x4 ConvT(3), Tanh | 64x64x3 | 2 |
| **Layers** | **Out Size** | **Stride** |
| Input: $z$ | 1x1x128 | - |
| 8x8 ConvT(ngf x 8), LReLU | 8x8x(ngf x 8) | 1 |
| 4x4 ConvT(ngf x 4), LReLU | 16x16x(ngf x 4) | 2 |
| 4x4 ConvT(ngf x 2), LReLU | 32x32x(ngf x 2) | 2 |
| 3x3 ConvT(3), Tanh | 32x32x3 | 1 |
| **Layers** | **Out Size** | **Stride** |
| Input: $z$ | 1x1x128 | - |
| 4x4 ConvT(ngf x 16), LReLU | 4x4x(ngf x 16) | 1 |
| 4x4 ConvT(ngf x 8), LReLU | 8x8x(ngf x 8) | 2 |
| 4x4 ConvT(ngf x 4), LReLU | 16x16x(ngf x 4) | 2 |
| 4x4 ConvT(ngf x 4), LReLU | 32x32x(ngf x 4) | 2 |
| 4x4 ConvT(ngf x 2), LReLU | 64x64x(ngf x 2) | 2 |
| 4x4 ConvT(ngf x 1), LReLU | 128x128x(ngfx1) | 2 |
| 4x4 ConvT(3), Tanh | 256x256x3 | 2 |
| **Layers** | **Out Size** | **Stride** |
| Input: $z$ | 1x1x8 | - |
| 7x7 ConvT(ngf x 8), LReLU | 7x7x(ngf x 8) | 1 |
| 4x4 ConvT(ngf x 4), LReLU | 14x14x(ngf x 4) | 2 |
| 4x4 ConvT(ngf x 2), LReLU | 28x28x(ngf x 2) | 2 |
| 3x3 ConvT(1), Tanh | 28x28x1 | 1 |

Table S3: **Network structure of the denoising diffusion network.** a) We use the sinusoidal embedding to embed the time index as in [25, 8]. b) We use the learned fourier feature module [75] to embed the input $\boldsymbol{z}$. c) The merged time embedding and context embedding is used to produce a pair of bias and scale terms to shift and scale the embedding of input $\boldsymbol{z}$. nz is the input dimension, as in Table S1. nemb is the dimension of image embedding as in Table S4.

| Layers | Out size | Note |
|---|---|---|
| **Time Embedding** | | |
| Input: $t$ | 1 | time index |
| Sin. emb.[a] | 128 | |
| Linear, SiLU | 128 | |
| Linear | 128 | |
| **Input Embedding** | | |
| Input: $\boldsymbol{z}$ | nz | |
| Fr. emb.[b] | 2x(nz) | Fourier feature |
| **Basic Block** | | |
| Input: $\boldsymbol{z}, \boldsymbol{z}_{\text{ctx}}, \boldsymbol{z}_t$ | nzf, nemb, 128 | $\boldsymbol{z}$, ctx. and t emb. |
| Cat, SiLU | nemb + 128 | merge ctx. & t emb. |
| Linear, SiLU | nout | |
| Linear | nout | Input emb. |
| Scale-shift[c] | nout | scale-shift $\boldsymbol{z}$ emb. w/ merged emb. |
| Add $\boldsymbol{z}$ | nout | skip connection from $\boldsymbol{z}$ |
| **Denoising Diffusion Network** | | |
| Input: $\boldsymbol{z}, \boldsymbol{z}_{\text{ctx}}, t$ | nz, nemb, 128 | Input |
| Embedding | 2x(nz), 128 | Input & t emb. |
| Basic Block | 128 | Encoding |
| Basic Block | 256 | |
| Basic Block | 256 | |
| Basic Block | 256 | Intermediate |
| Basic Block | 256 | Cat & Decoding |
| Basic Block | 128 | |
| Basic Block | nz | Output |

**Hyperparameters and training details**     As mentioned in the main text, for the posterior and prior DAMC samplers, we set the number of diffusion steps to 100. The number of iterations in Eq. (8) is set to $M = 6$ for the experiments. The LD runs $T = 30$ and $T = 60$ iterations for posterior and prior updates during training with a step size of $s = 0.1$. For test time sampling from $\mathcal{K}_{T, \boldsymbol{z}_i | \boldsymbol{x}_i} q_{\boldsymbol{\phi}_k}(\boldsymbol{z}_i | \boldsymbol{x}_i)$, we set $T = 10$ for the additional LD. For test time prior sampling of LEBM with LD, we follow [22, 41] and set $T = 100$. To further stabilize the training procedure, we i) perform gradient clipping by setting the maximal gradient norm as 100, ii) use a separate target diffusion network which is the EMA of the current diffusion network to initialize the prior and posterior updates and iii) add noise-initialized prior samples for the prior updates. These set-ups are identical across different datasets.

The parameters of all the networks are initialized with the default pytorch methods [77]. We use the Adam optimizer [78] with $\beta_1 = 0.5$ and $\beta_2 = 0.999$ to train the generator network and the energy score network. We use the AdamW optimizer [79] with $\beta_1 = 0.5$, $\beta_2 = 0.999$ and `weight_decay=1e-4` to train the diffusion network. The initial learning rates of the generator and diffusion networks are `2e-4`, and `1e-4` for the energy score network. The learning rates are decayed with a factor of 0.99 every 1K training iterations, with a minimum learning rate of `1e-5`. We run the experiments on a A6000 GPU with the batch size of 128. For GAN inversion, we reduce the batch size to 64. Training typically converges within 200K iterations on all the datasets.

Table S4: **Network structures of the encoder networks** used for the SVHN, CelebA, CIFAR-10, CelebA-HQ and MNIST (from top to bottom) datasets. For GAN inversion, the encoder network structure is the same as in [18]. Conv($n$)Norm indicates a convolutional operation with $n$ output channels followed by the Instance Normalization [76]. We use `nif=64` and `nemb=1024` for all the datasets. LReLU indicates the Leaky-ReLU activation function. The slope in Leaky ReLU is set to be 0.2.

| Layers | Out Size | Stride |
|---|---|---|
| Input: $x$ | 32x32x3 | - |
| 3x3 Conv(nif x 1)Norm, LReLU | 32x32x(nif x 1) | 1 |
| 4x4 Conv(nif x 2)Norm, LReLU | 16x16x(nif x 2) | 2 |
| 4x4 Conv(nif x 4)Norm, LReLU | 8x8x(nif x 4) | 2 |
| 4x4 Conv(nif x 8)Norm, LReLU | 4x4x(nif x 8) | 2 |
| 4x4 Conv(nemb)Norm, LReLU | 1x1x(nemb) | 1 |
| **Layers** | **Out Size** | **Stride** |
| Input: $x$ | 64x64x3 | - |
| 3x3 Conv(nif x 1)Norm, LReLU | 64x64x(nif x 1) | 1 |
| 4x4 Conv(nif x 2)Norm, LReLU | 32x32x(nif x 2) | 2 |
| 4x4 Conv(nif x 4)Norm, LReLU | 16x16x(nif x 4) | 2 |
| 4x4 Conv(nif x 8)Norm, LReLU | 8x8x(nif x 8) | 2 |
| 4x4 Conv(nif x 8)Norm, LReLU | 4x4x(nif x 8) | 2 |
| 4x4 Conv(nemb)Norm, LReLU | 1x1x(nemb) | 1 |
| **Layers** | **Out Size** | **Stride** |
| Input: $x$ | 32x32x3 | - |
| 3x3 Conv(nif x 1)Norm, LReLU | 32x32x(nif x 1) | 1 |
| 4x4 Conv(nif x 2)Norm, LReLU | 16x16x(nif x 2) | 2 |
| 4x4 Conv(nif x 4)Norm, LReLU | 8x8x(nif x 4) | 2 |
| 4x4 Conv(nif x 8)Norm, LReLU | 4x4x(nif x 8) | 2 |
| 4x4 Conv(nemb)Norm, LReLU | 1x1x(nemb) | 1 |
| **Layers** | **Out Size** | **Stride** |
| Input: $x$ | 256x256x3 | - |
| 3x3 Conv(nif x 1)Norm, LReLU | 256x256x(nif x 1) | 1 |
| 4x4 Conv(nif x 2)Norm, LReLU | 128x128x(nif x 2) | 2 |
| 4x4 Conv(nif x 4)Norm, LReLU | 64x64x(nif x 4) | 2 |
| 4x4 Conv(nif x 4)Norm, LReLU | 32x32x(nif x 4) | 2 |
| 4x4 Conv(nif x 8)Norm, LReLU | 16x16x(nif x 8) | 2 |
| 4x4 Conv(nif x 8)Norm, LReLU | 8x8x(nif x 8) | 2 |
| 4x4 Conv(nif x 8)Norm, LReLU | 4x4x(nif x 8) | 2 |
| 4x4 Conv(nemb)Norm, LReLU | 1x1x(nemb) | 1 |
| **Layers** | **Out Size** | **Stride** |
| Input: $x$ | 28x28x3 | - |
| 3x3 Conv(nif x 1)Norm, LReLU | 28x28x(nif x 1) | 1 |
| 4x4 Conv(nif x 2)Norm, LReLU | 14x14x(nif x 2) | 2 |
| 4x4 Conv(nif x 4)Norm, LReLU | 7x7x(nif x 4) | 2 |
| 4x4 Conv(nif x 8)Norm, LReLU | 3x3x(nif x 8) | 2 |
| 3x3 Conv(nemb)Norm, LReLU | 1x1x(nemb) | 1 |

# C  Pytorch-style Pseudocode

We provide pytorch-style pseudocode to help understand the proposed method. We denote the generator network as `G`, the energy score network as `E` and the diffusion network as `Q`. The first page sketches the prior and posterior sampling process. The second page outlines the learning procedure.

Listing 1: Prior and posterior LD sampling.

```python
def sample_langevin_prior_z(z, netE):
    s = step_size

    for i in range(n_steps):
        en = netE(z).sum()
        z_norm = 1.0 / 2.0 * torch.sum(z**2)
        z_grad = torch.autograd.grad(en + z_norm, z)[0]
        w = torch.randn_like(z)

        # Prior LD Update
        z.data = z.data - 0.5 * (s ** 2) * z_grad + s * w

    return z.detach()

def sample_langevin_posterior_z(z, x, netG, netE):
    s = step_size
    sigma_inv = 1.0 / (2.0 * sigma ** 2)

    for i in range(n_steps):
        x_hat = netG(z)
        g_log_lkhd = sigma_inv * torch.sum((x_hat - x) ** 2)

        z_n = 1.0 / 2.0 * torch.sum(z**2)
        en = netE(z).sum()

        total_en = g_log_lkhd + en + z_n
        z_grad = torch.autograd.grad(total_en, z)[0]
        w = torch.randn_like(z)

        # Posterior LD Update
        z.data = z.data - 0.5 * (s ** 2) * z_grad + s * w
    return z.detach()
```

Listing 2: Learning LEBM with DAMC.

```python
for x in dataset:
    # mask for unconditional learning of DAMC
    z_mask_prob = torch.rand((len(x),), device=x.device)
    z_mask = torch.ones(len(x), device=x.device)
    z_mask[z_mask_prob < 0.2] = 0.0
    z_mask = z_mask.unsqueeze(-1)

    # draw DAMC samples
    z0 = Q(x)
    zk_pos, zk_neg = z0.detach().clone(), z0.detach().clone()

    # prior and posterior updates
    zk_pos = sample_langevin_posterior_z(
            z=zk_pos, x=x, netG=G, netE=E)
    zk_neg = sample_langevin_prior_z(
            z=torch.cat(
            [zk_neg, torch.randn_like(zk_neg)], dim=0),
            netE=E)

    # update Q
    for __ in range(6):
        Q_optimizer.zero_grad()
        Q_loss = Q.calculate_loss(
        x=x, z=zk_pos, mask=z_mask).mean()
        Q_loss.backward()
        Q_optimizer.step()

    # update G
    G_optimizer.zero_grad()
    x_hat = G(zk_pos)
    g_loss = torch.sum((x_hat - x) ** 2, dim=[1,2,3]).mean()
    g_loss.backward()
    G_optimizer.step()

    # update E
    E_optimizer.zero_grad()
    e_pos, e_neg = E(zk_pos), E(zk_neg)
    E_loss = e_pos.mean() - e_neg.mean()
    E_loss.backward()
    E_optimizer.step()
```

# D   Dataset and Experiment Settings

**Datasets**   We include the following datasets to study our method: SVHN ($32 \times 32 \times 3$), CIFAR-10 ($32 \times 32 \times 3$), CelebA ($64 \times 64 \times 3$), CeleAMask-HQ (256 x 256 x 3) and MNIST (28 x 28 x 1). Following Pang et al. [22], we use the full training set of SVHN (73,257) and CIFAR-10 (50,000), and take 40,000 samples of CelebA as the training data. We take 29,500 samples from the CelebAMask-HQ dataset as the training data, and test the model on 500 held-out samples. For anomaly detection on MNIST dataset, we follow the experimental settings in [22, 41, 55, 56] and use 80% of the in-domain data to train the model. The images are scaled to $[-1, 1]$ and randomly horizontally flipped with a prob. of .5 for training.

**GAN inversion settings**   We attempt to use the DAMC sampler for GAN version on the FFHQ (256 x 256 x 3) and LSUN-Tower (256 x 256 x 3) datasets. We take 69,500 samples from the CelebAMask-HQ dataset as the training data, and use the held-out 500 samples for testing. We follow the default data splits of the LSUN dataset.

For the LEBM-based inversion method, we train a LEBM in the 14 x 512 = 7168 dim. latent space. During training, we add $l_2$-regularization on the energy score of LEBM to stabilize training, as suggested in [59]. For the DAMC sampler and the encoder-based inversion method [16], we generate initial posterior samples of the training data using [18] and train the DAMC sampler and the encoder-based method for 5K iterations using these samples as a warm-up step. The encoder-based method is trained by minimizing the $l_2$ distance between the encoder output and the target samples. After that, these two methods are trained with the default learning algorithms.

# E  Additional Qualitative Results

## E.1  Toy Examples

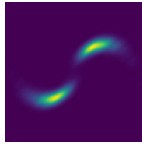

Figure S1: **The 2-arm pinwheel-shaped prior distribution used in the toy example.**

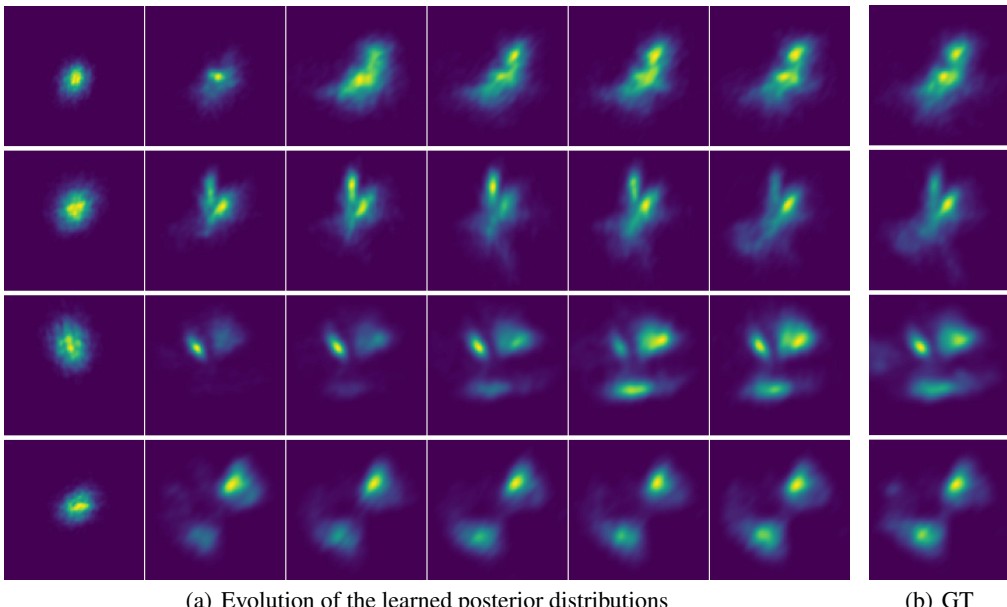

(a) Evolution of the learned posterior distributions       (b) GT

Figure S2: **Evolution of the posterior distributions learned by the DAMC sampler.** In each row, we display from left to right the evolution of the learned posterior distributions from the DAMC sampler through training iterations. The last column shows the corresponding ground-truth posterior distribution obtained by running 1K-3K steps of Langevin Dynamics until convergence for posterior sampling.

As the proof-of-concept toy examples, we implement the neural likelihood experiments following the same set-up mentioned in Section 5.1 and B.1 in [80]. We choose to use a more complex prior distribution, i.e., 2-arm pinwheel-shaped prior distribution (shown in Fig. S1) instead of a standard normal one to make sure that the true posterior distributions are multimodal. The ground-truth posterior distributions are obtained by performing long-run LD sampling until convergence. We visualize the convergence trajectory of the posterior distributions learned by our model to the ground-truth ones. In Fig. S2, we can see that our model can faithfully reproduce the ground-truth distributions with sufficient training iterations, which indicates that the learned sampler successfully amortizes the long-run sampling chain with the length of $1,000$-$3,000$ iterations.

## E.2 Generation

We provide additional generated samples from our models trained on SVHN (Fig. S3), CelebA (Fig. S4), CIFAR-10 (Fig. S5) and CelebA-HQ (Fig. S6).

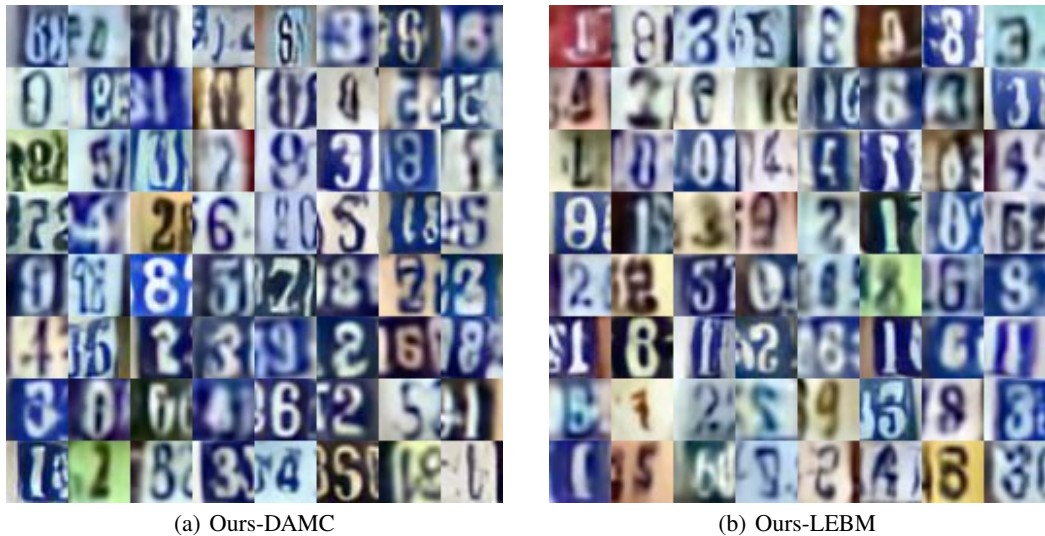

(a) Ours-DAMC                    (b) Ours-LEBM

Figure S3: **Samples generated from the DAMC sampler and LEBM** trained on the SVHN dataset.

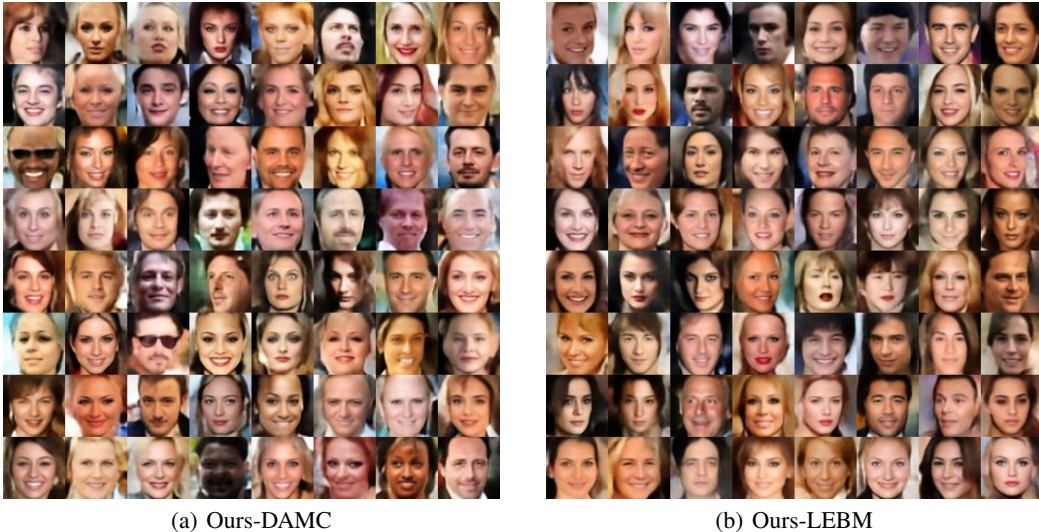

(a) Ours-DAMC                    (b) Ours-LEBM

Figure S4: **Samples generated from the DAMC sampler and LEBM** trained on the CelebA dataset.

## E.3 Reconstruction

We provide qualitative examples about the reconstruction results from our models trained on SVHN (Fig. S7), CelebA (Fig. S8), CIFAR-10 (Fig. S9) and CelebA-HQ (Fig. S10). Observed images are sampled from the testing set unseen during training.

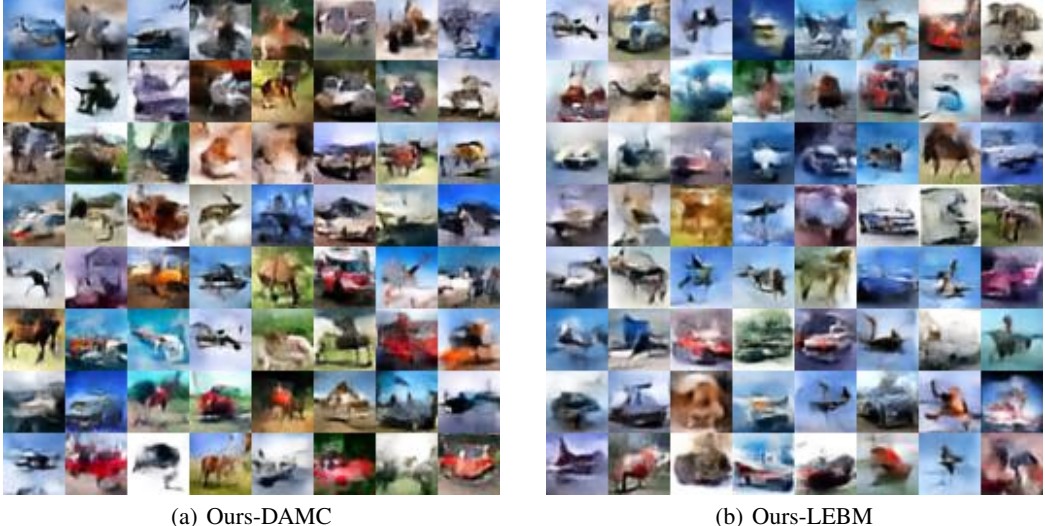

| | |
|---|---|
| (a) Ours-DAMC | (b) Ours-LEBM |

Figure S5: **Samples generated from the DAMC sampler and LEBM** trained on the CIFAR-10 dataset.

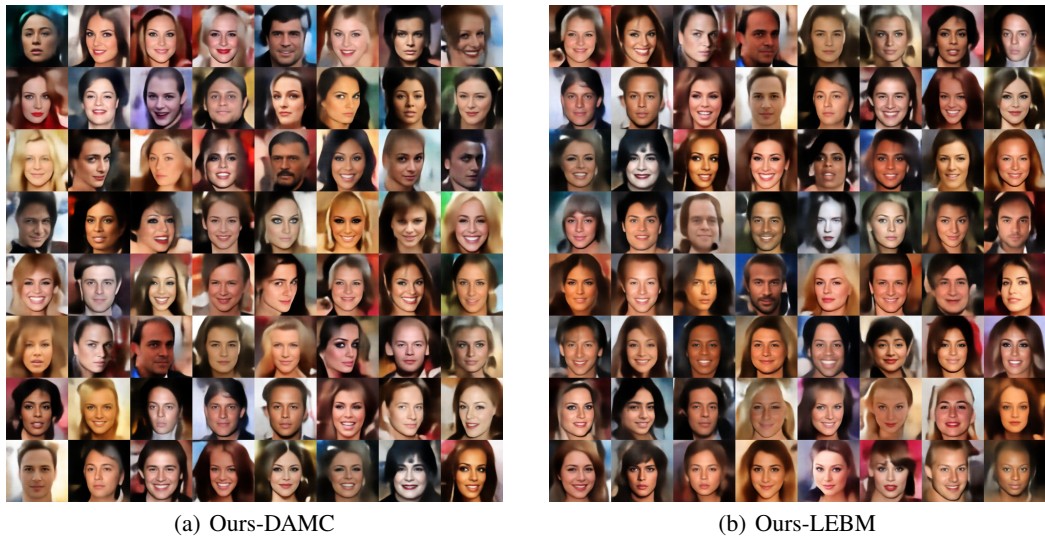

| | |
|---|---|
| (a) Ours-DAMC | (b) Ours-LEBM |

Figure S6: **Samples generated from the DAMC sampler and LEBM** trained on the CelebA-HQ dataset.

## E.4 Visualization of Transitions

We provide additional visualization results of LD transitions initialized from $\mathcal{N}(0, \mathbf{I}_d)$ on SVHN (Fig. S11) and CIFAR-10 datasets (Fig. S12). For the 200-step set-up, we can see that the generation quality quickly improves by exploring the local modes with LD. For the 2500-step long-run set-up, we can see that the LD produces consistently valid results without the oversaturating issue of the long-run chain samples.

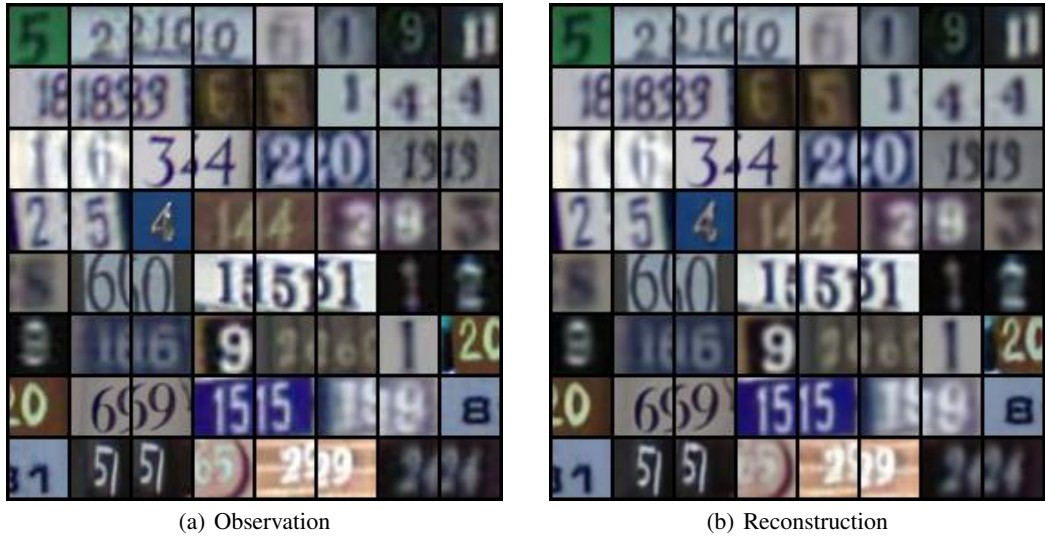

(a) Observation        (b) Reconstruction

Figure S7: **Reconstructed samples from the posterior DAMC sampler** trained on the SVHN dataset.

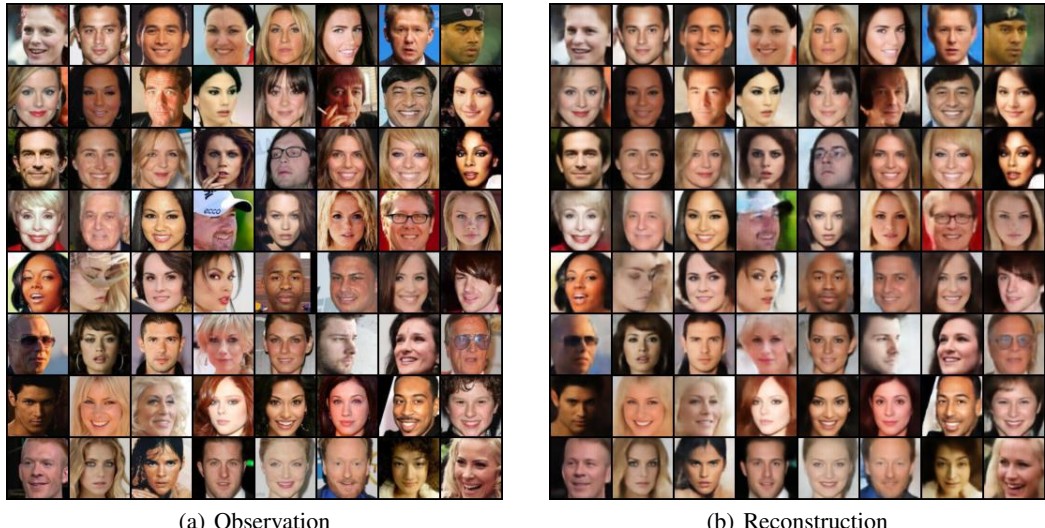

(a) Observation        (b) Reconstruction

Figure S8: **Reconstructed samples from the posterior DAMC sampler** trained on the CelebA dataset.

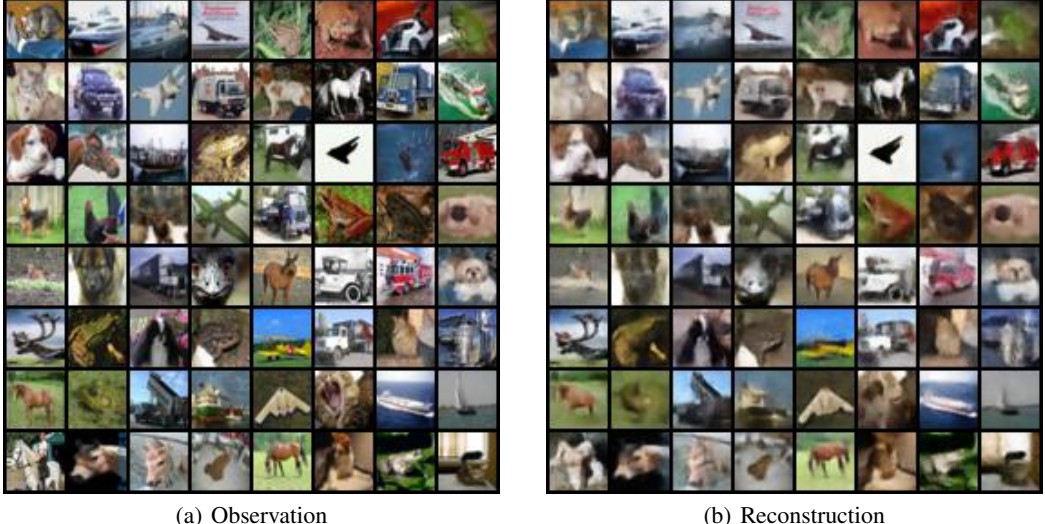

(a) Observation                                    (b) Reconstruction

Figure S9: **Reconstructed samples from the posterior DAMC sampler** trained on the CIFAR-10 dataset.

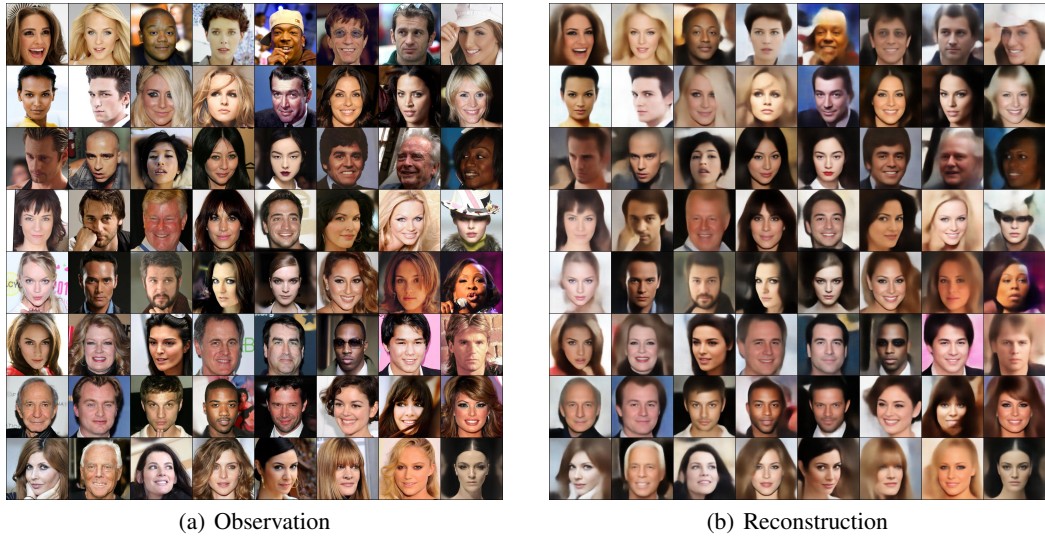

(a) Observation                                    (b) Reconstruction

Figure S10: **Reconstructed samples from the posterior DAMC sampler** trained on the CelebA-HQ dataset.

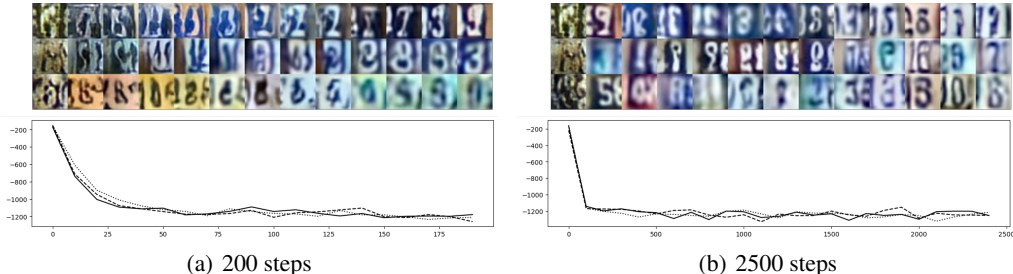

(a) 200 steps                                    (b) 2500 steps

Figure S11: **Transition of Markov chains initialized from $\mathcal{N}(0, \mathbf{I}_d)$ towards $p_{\boldsymbol{\alpha}}(\boldsymbol{z})$ on SVHN.** We present results by running LD for 200 and 2500 steps. In each sub-figure, the top panel displays the trajectory in the data space uniformly sampled along the chain. The bottom panel shows the energy score $f_{\boldsymbol{\alpha}}(\boldsymbol{z})$ over the iterations.

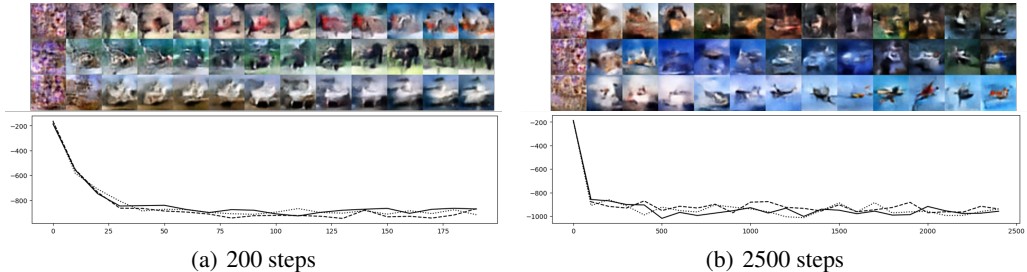

(a) 200 steps            (b) 2500 steps

Figure S12: **Transition of Markov chains initialized from** $\mathcal{N}(0, \mathbf{I}_d)$ **towards** $p_{\boldsymbol{\alpha}}(\boldsymbol{z})$ **on CIFAR-10.** We present results by running LD for 200 and 2500 steps. In each sub-figure, the top panel displays the trajectory in the data space uniformly sampled along the chain. The bottom panel shows the energy score $f_{\boldsymbol{\alpha}}(\boldsymbol{z})$ over the iterations.

# F Additional Quantitative Results

## F.1 Learning DAMC in the Latent Space of Other DLVMs

We have compared our method with directly training diffusion models in the latent space. In the NALR-LEBM column in Table 4, we compare our method with learning a diffusion model in the pre-trained energy-based latent space on CIFAR-10 dataset. We further add experiments of learning this model in the pre-trained VAE [1] and ABP [51] model latent spaces summarized in Table S5. We can see that our method greatly outperforms these baselines using the same network architectures.

Table S5: **Further baseline results for learning DAMC on CIFAR-10 dataset.** These models are implemented using the same encoder and decoder network architectures for fair comparison

|      | VAE    | ABP   | NALR  | Ours    |
|------|--------|-------|-------|---------|
| FID  | 102.54 | 69.93 | 64.38 | **57.52** |
| MSE  | 0.036  | 0.017 | 0.016 | **0.015** |

## F.2 Further Ablation Studies

We conduct more ablation studies to inspect the impact of some hyper-parameters. For prior LD sampling, we follow [22] and set the sampling step as 60 throughout the experiments for fair comparison. We further add experiments on CIFAR-10 for training our model with different posterior LD sampling steps T and with different capacities of the amortizer $q_\phi$, summarized in Table S6. We observe that larger posterior sampling steps and larger model capacities for training brings marginal improvement compared with our default set-up. Less sampling steps and lower model capacities, however, may have negative impact on the performances.

| (a) Posterior sampling steps T | | | | (b) Model capacity of $q_\phi$ | | | | | |
|------|--------|--------|--------|------|--------|--------|--------|--------|--------|
|      | T=10   | T=30   | T=50   |      | f=1/4  | f=1/2  | f=1    | f=2    | f=4    |
| FID  | 74.20  | **57.72** | 57.03 | FID  | 116.28 | 80.28  | **57.52** | 57.82  | 57.56  |
| MSE  | 0.016  | **0.015** | 0.015 | MSE  | 0.017  | 0.016  | **0.015** | 0.015  | 0.015  |

Table S6: **Ablation Studies for the choice of Langevin steps and model capacity**. We highlight the results of our set-up reported in the main text. T is the posterior sampling step. f stands for the factor for the model capacity, *e.g.*, f=2 means 2x the size of the original model.

# G  Further Discussion

## G.1  Limitations

We mentioned in the main text that one potential disadvantage of our method is its parameter ineffi­ciency for introducing an extra DDPM. Although fortunately, our models are in the latent space so the network is lightweight. To be specific, on SVHN, CelebA, CIFAR-10 and CelebA-HQ datasets the number of parameters in the diffusion network is around $10\%$ of those in the generator.

Another issue is the time efficiency. We mentioned in the main text that the time efficiency for sampling is competitive. With the batch size of $64$, on these datasets the DAMC prior sampling takes $0.3s$, while $100$ steps of short-run LD with LEBM takes $0.2s$. The DAMC posterior sampling takes $1.0s$, while LEBM takes $8.0s$. However, during training we need to run 30 steps of posterior LD sampling and 60 steps of prior LD sampling in each training iteration. We observe that the proposed learning method takes 15.2 minutes per training epoch, while the short-run LD-based learning method takes 14.8 minutes per epoch. These methods are slower than the VAE-base method, which takes 5.5 minutes for an training epoch. We can see that the time efficiency for training is generally bottlenecked by the LD sampling process, and could be improved in future works.

## G.2  Broader Impacts

Generative models could be misused for disinformation or faking profiles. Our work focuses on the learning algorithm of energy-based prior model. Though we consider our work to be foundational and not tied to particular applications or deployments, it is possible that more powerful energy-based generative models augmented with this method may be used maliciously. Work on the reliable detection of synthetic content could be important to address such harms from generative models.

