# Supplementary of Learning Energy-Based Prior Model with Diffusion-Amortized MCMC

## Contents

Preprint. Under review.

# A Related Work

**Energy-based prior model**  Energy-Based Models (EBMs) [1–5] play an important role in generative modeling. Pang et al. [6] propose to learn an EBM as a prior model in the latent space of deep latent variable models; it greatly improves the model expressivity over those with non-informative priors and brings strong performance on downstream tasks, *e.g.*, image segmentation, text modeling, molecule generation, and trajectory prediction [7–10]. However, learning both EBMs and latent space EBMs require MCMC sampling to estimate the learning gradients, which requires a large amount of iterations to converge when the target distributions are high-dimensional or highly multi-modal. Typical choices of sampling with non-convergent short-run MCMC [2] in practice can lead to poor generation quality, malformed energy landscapes [2, 8, 11], biased estimation of the model parameter and instability in training [3–5, 11]. In this work, we consider learning valid amortization of the potentially long-run MCMC for learning energy-based priors; the proposed model shows reliable sampling quality in practice.

**Denoising diffusion probabilistic model**  Denoising Diffusion Probabilistic Models (DDPMs) [12–15], originating from Sohl-Dickstein et al. [12], learn the generative process by recovering the observed data from a sequence of noise-perturbed versions of the data. The learning objective can be viewed as a variant of the denoising score matching objective [16]. As pointed out in [12, 13], the sampling procedure of DDPM with $\epsilon$-prediction parametrization resembles Langevin Dynamics (LD) of an EBM; $\epsilon$ (predicted noise) plays a similar role to the gradient of the log density [13]. To be specific, learning a DDPM with $\epsilon$-prediction parameterization is equivalent to fitting the finite-time marginal of a sampling chain resembling annealed Langevin dynamics [13–15]. Inspired by this connection, we propose to amortize the long-run MCMC in learning energy-based prior by iteratively distilling the short-run sampling chain segments with a diffusion-based sampler. We provide empirical and theoretical evidence that the resulting sampler is a valid long-run chain sampler.