# OpenReview forum: "Learning Energy-Based Prior Model with Diffusion-Amortized MCMC"
_NeurIPS.cc/2023/Conference — NeurIPS 2023 poster_

### Official Review · Reviewer_rkuE · 2023-07-06

**Soundness:** 3 good
**Presentation:** 2 fair
**Contribution:** 2 fair
**Rating:** 3
**Confidence:** 2

**Summary:**

The paper presents a new training and sampling procedures for learning energy based generative models.  The method is compared to earlier (few-step) MCMC based approaches and to diffusion models.  The procedure is evaluated on computer vision tasks.

**Strengths:**

The paper presents an original algorithm, with a clear methodological description, and a large collection of benchmarking results.

**Weaknesses:**

The paper could make clearer for the general audience why additional energy based modeling methodology is needed.  As a researcher who has been working extensively on diffusion models for the past two years, but has not worked with (other) energy based models in approaches in recent years additional context in goals of this direction and the limitations of previous approaches would be helpful.

For example, the authors comment that EBMs are "closely related to" DDPMs.  Which do not suffer from the "sampling issue" associated with "non-convergent short-run MCMC" used for EBMs.   Is there something that can be accomplished with EBMs that is not solved by DDPMs that motivates their continued study?

The paper is also unclear with respect to the claim for "theoretical evidence that the learned amortization of MCMC is a valid long-run MCMC sampler".  Could the authors make this theoretical claim more explicit (e.g. as a proposition or theorem)?

**Questions:**

No further questions.

**Limitations:**

Limitations are adequately addressed.

---

> ### Author Rebuttal · Authors · 2023-08-09
>
> ### Thank you for your detailed comments
> We sincerely thank you for your time and detailed comments. Below, we provide point-to-point responses to hopefully address the concerns you have.
>
> - > Additional context in goals of this direction and the limitations of previous approaches would be helpful. Is there something that can be accomplished with EBMs that is not solved by DDPMs that motivates their continued study?
>     - Thank you for bringing this matter to our attention. We have generally introduced the background and related works of this direction in sec. 1 and 2 in the main text and sec. A in the appendix. We will make sure to more explicitly discuss these related works in revision.
>     - An example of "something that can be accomplished with EBMs that is not solved by DDPMs" can be found in [1] with a brief but insightful discussion in sec. 4.2 of [1]. For tasks that requires explicit likelihood function, e.g., compositional generation, the energy-based parameterization (EBM) is preferred over epsilon-parameterization (DDPM) for its flexibility in modeling. This energy-based parametrization enables the use of more accurate samplers for greatly improved sample quality and convergence in the compositional generation task compared with DDPM.
>     - More broadly speaking, there are certain distributions where the epsilon-parameterization cannot generate decent samples [2]. With the standard epsilon-parameterization one can only utilize unadjusted samplers, which perform well in practice. But for target distributions such as those with lighter-than-Gaussian tails [2], the unadjusted sampler chain is transient and may not produce desired samples. Our method in this submission learns both the latent space EBM and its corresponding amortized sampler, thus combining the strengths of both sides, i.e., a well-learned explicit (unnormalized) density and a strong sampler that greatly mitigate the sampling issue.
>
> - > Could the authors make the theoretical claim more explicit (e.g. as a proposition or theorem)?
>     - Thank you for pointing this out. We will consider revising our paper accordingly.
>
> We hope our answers could further address the reviewer’s concerns of this work. If you have any additional questions/comments/concerns, please feel free to let us know here.
>
> [1] Du et al. Reduce, Reuse, Recycle: Compositional Generation with Energy-Based Diffusion Models and MCMC. ICML 2023.
>
> [2] Roberts and Tweedie. Exponential convergence of Langevin distributions and their discrete approximations. Bernoulli, pp. 341–363, 1996.

---

> > ### Comment · Reviewer_rkuE · 2023-08-11
> > **Thank you and follow-up question**
> >
> > Thank you for your replies.
> >
> > To follow-up on your question, in the context of your paper is a DDPM an example of an energy based model once (re-)parameterized in terms of its implied approximation to the gradient of log-densities?  If so statement (w.r.t. reference [1] above) this seems to be more of a statement about what parameterization of diffusion models than energy based models as a distinct class of models.
> >
> > As the motivation for why this class of models is considered and how it might hope to address limitations of existing diffusion base generative models remains unclear to me, I maintain my score and low-confidence.

---

> > > ### Author Response · Authors · 2023-08-12
> > > **Thanks for your reply**
> > >
> > > Thank you for your prompt response.
> > >
> > > We would like to point out that although DDPM estimates the gradient of the log density of *noise-perturbed versions of the target density*, this gradient estimate **does not** translate to reliable estimate of the *target density (without noise)* via re-parametrization. For a given noise level $\sigma > 0$, the denoising score matching or DDPM objective is not a consistent objective for learning the underlying target density without noise. Annealing the noise level $\sigma \to 0$ can mitigate this problem, but the gradient estimation is only reliable in the immediate vicinity of the modes of the target distribution, where the density is high. For low-density regions of the target distribution, the objective may not have enough evidence to estimate score functions accurately, due to the lack of samples [1].
> > >
> > > In contrast, EBM is typically learned through Maximum Likelihood Estimation, which theoretically is the most accurate estimate in terms of asymptotic variance [2]. This formulation together with its learning algorithm produces much more reliable explicit (unnormalized) density estimation. From this point of view, these two models are different.
> > >
> > > In our previous response, we discussed the importance of energy-based modeling, i.e., learning the density of target distribution explicitly. We hope the key differences between EBM and DDPM mentioned above in this response further clarifies the necessity of learning the explicit density of the target distribution.
> > >
> > > Specifically in the set-up of our submission, we care about modeling the latent space. The latent space EBM provides explicit prior probability density $p_\alpha(z)$ for modeling latent variables $z$, and consequently defines explicit posterior density $p_\theta(z|x) \propto p_\alpha(z)p_\beta(x|z)$ for posterior inference upon which we build our approximate MLE learning algorithm. Simply plugging a DDPM into the latent space does not provide us with this well-defined MLE learning framework and could lead to problematic learning algorithms.
> > >
> > > Finally, we would like to kindly refer to [3] for a comprehensive discussion about EBM and its connection with DDPM. We hope our response helps to address your concerns and explains why latent space EBM is considered in this work. Please feel free to let us know if you have any additional questions/comments/concerns.
> > >
> > > [1] Yang Song, and Stefano Ermon. Generative Modeling by Estimating Gradients of the Data Distribution. NeurIPS 2019.
> > >
> > > [2] Peter J. Bickel and Kjell A. Doksum, Mathematical Statistics: Basic Ideas and Selected Topics, Volume I.
> > >
> > > [3] Yang Song and Diederik P. Kingma. How to Train Your Energy-Based Models, arXiv:2101.03288

---

> > > > ### Comment · Reviewer_rkuE · 2023-08-16
> > > > **Thank you for your reply.**
> > > >
> > > > I appreciate the replies but remain uncertain in my assessment.  Thank you.

---

### Official Review · Reviewer_hz9t · 2023-07-07

**Soundness:** 3 good
**Presentation:** 3 good
**Contribution:** 2 fair
**Rating:** 4
**Confidence:** 4

**Summary:**

This paper proposes the DAMC sampler (Diffusion-Amortized MCMC) and develops a new learning algorithm for LEBM (Latent-space Energy-Based Model) based on it. Theoretical and empirical evidences are provided for the effectiveness of our method.


**Strengths:**

The paper is generally well-written. The idea of amortizing the LD with DDPM in learning the LEBM seems to be new.


**Weaknesses:**

The basic idea of using an auxiliary model to amortize the LD in learning energy-based models has been well known and explored in the literatures, e.g. in [a,b] to name a few. Methodology connection and experiment comparison with those methods are needed.

By looking at FID on CIFAR-10 (Table 1), the performance of the proposed method (FID 57.72) is far behind previous methods [a,b] (33.61, 20.9).

In the current literature, long-run MCMC analysis is typically conducted over a range of 10,000 to 100,000 Langevian Dynamics iterations. However, in this paper (Figure 4), the authors perform a much shorter 2,500 Langevian dynamics update, which is considerably less extensive compared to other studies in the field.

[a] Cooperative Training of Descriptor and Generator Networks, arXiv:1609.09408v3
[b] Learning Neural Random Fields with Inclusive Auxiliary Generators, arXiv:1806.00271v4


**Questions:**

see above.

**Limitations:**

see above.

---

> ### Author Rebuttal · Authors · 2023-08-09
>
> ### Thank you for your insightful comments
> We sincerely thank you for your time and thoughtful comments! Below, we provide point-to-point replies to your comments that hopefully would address the concerns you have.
>
> - > Methodology connection and experiment comparison with existing methods using an auxiliary model to amortize the LD, e.g., [a] and [b] are needed. The FID scores on CIFAR-10 are far behind [a] and [b]
>     - Thank you for pointing us to these interesting works! We have generally introduced the amortized MCMC methods including those using an auxiliary model in sec. A in the appendix. We will make sure to explicitly connect and discuss these methods including [a,b] in revision.
>     - We would like to point out that both [a] and [b] trains the auxiliary models and energy-based models in the pixel space, which is essentially different from our set-up. In this paper, we focus on learning the energy-based prior and its corresponding posterior and prior sampler in the latent space of generative models. One of the key differences is that model design for learning the latent space samplers requires non-trivial extra efforts, since the latent space is ever-changing during training unlike the data space; delicate designs are often needed to balance the learning of this amortized model and other jointly learned models for stable training, if we want to employ these pixel space methods. In this work, we follow the method of [c] to train a single neural network to parameterize both the prior and posterior sampling models to deal with this problem.
>     - In addition, as mentioned in sec. 3.3 L184, we use the same network architectures as in previous works [d,e] for fair comparison with these methods. To be specific, as shown in sec. C in appendix the generator network used in this work for the CIFAR-10 experiment uses 5 layers of transposed convolution, while the generator network in [b] uses a ResBlock-based deeper generator. The FID scores are therefore not directly comparable between ours and those reported in [a] and [b]. We hope to explore more advanced architectures to address these issues in future work.
>
> - > In the current literature, long-run MCMC analysis is typically conducted over a range of 10,000 to 100,000 Langevian Dynamics iterations.
>     - Thank you for bringing this matter to our attention! We checked the performance of our model for longer LD chain on CIFAR-10, SVHN and CelebA64 datasets by calculating the FID scores of long-run samples. The results are summarized below. We can see that our model could produce consist results with long-run chains.
> |Steps|100|200|2500|10000|100000
> |:--|:--:|:--:|:--:|:--:|:--:|
> |CIFAR-10|60.89|60.73|61.20|60.76|61.20|
> |SVHN|21.17|20.91|20.07|20.68|20.71|
> |CelebA64|35.67|35.39|35.40|35.29|35.17|
>
> Thank you again for providing constructive and thoughtful feedback on our submission. If you have any additional questions/comments/concerns, please feel free to let us know here. **Otherwise, we would appreciate it if you would consider raising your rating of this submission**.
>
> [a] Cooperative Training of Descriptor and Generator Networks, arXiv:1609.09408v3
>
> [b] Learning Neural Random Fields with Inclusive Auxiliary Generators, arXiv:1806.00271v4
>
> [c] Classifier-Free Diffusion Guidance. NeurIPS 2021 Workshop.
>
> [d] Learning latent space energy-based prior model. NeurIPS 2020.
>
> [e] Adaptive multi-stage density ratio estimation for learning latent space energy-based model. NeurIPS 2022.

---

> > ### Comment · Reviewer_hz9t · 2023-08-17
> > **Concern about the insufficiency of experimental results remains**
> >
> > --after reading feedback--
> >
> > Thanks for the feedback from the authors.
> >
> > The new result from long-run MCMC seems good.
> >
> > Learning the energy-based model in the latent space may require more delicate design than learning in the pixel space. But the basic math required is not so innovative.
> >
> > The authors refer their architectures to previous works [d,e], and I can see that the network architectures used in this work are weaker than those used in [a,b]. However, remarkably, the results in [a,b] dates back to 2016-2018. If the results in 2023 cannot show improvements over these classic ones, I'm not convinced that the claimed effectiveness of the new method in this work makes a real progress in learning EBMs. One may doubt whether the proposed method can work or not when using a deeper architecture. My main concern about the insufficiency of experimental results remains. I suggest that the authors show results using a bit deeper advanced network, which is not difficult for experiments. I think, such comparison would be really beneficial for the community to advance the EBM study.

---

> > > ### Author Response · Authors · 2023-08-20
> > > **Thank you for your reply**
> > >
> > > Thank you for your further response. We provide point-to-point replies below.
> > >
> > > - > If the results in 2023 cannot show improvements over these classic ones, the proposed method might not make a real progress in learning EBMs.
> > >
> > >     We respectfully disagree with this statement for the following reasons.
> > >     - **The proposed method and the mentioned previous methods work for different models in completely different set-ups.** We would like to continue our discussion in our previous response to further clarify that these methods are essentially different.
> > >
> > >         As mentioned in sec. 1 L16-28 in the main text, we specifically consider *learning an EBM in the latent space for latent variables $z$ (LEBM) as an informative prior* for the generator network in this work. Although our focus is also on learning an EBM, the LEBM stands on the generator to capture the regularity of its latent variables $z$, while the main generation process is done by the generator network $g$. The learning process is built upon the posterior inference of $z$ to formulate an EM-like approximate MLE learning algorithm. In this formulation, the generator is directly supervised by the reconstruction error of the observed data $x$.
> > >
> > >         In contrast, in the set-up of [a,b], the EBM resides explicitly in the data space for $x$ and serves as a refiner of the initial output of the generator network. The learning process does not involve inference of $z$ or modeling of the latent space. In [a], the refined output is further used as the training sample of the generator, while in [b] the generator is trained by optimizing the learned energy score function.
> > >
> > >         In sum, the key differences between our set-up and [a,b] include, at least, the following ones:
> > >         - Although we all consider learning EBMs, the roles of EBMs are completely different. In our work, the light-weight LEBM is used as a flexible yet powerful prior for the generator, while in [a,b] the EBMs are used as a refiner or the main generation model;  the learning processes in [a,b] do not involve inference of $z$ or modeling of the latent space.
> > >         - Although both our work and [a,b] involve learning the generator, we can see in [a,b] the generator is supervised by a different objective that incorporates the training signal from the data-space EBM, rather than directly optimizing the reconstruction error of the original observed data $x$, i.e., $\log p(x | z)$ in our set-up.
> > >
> > >         Therefore, we think the differences of FID scores of these methods **may not** serve as valid indicators, since the methods work for different models in completely different set-ups and are not directly comparable.
> > >
> > >
> > >     - **The proposed method and the mentioned previous methods are not direct competitors, but are actually complementary to each other.**
> > >     Following our discussion above, we would like to add that a more reasonable comparison would be incorporating the LEBM in the latent space of the generator used in [a,b], and jointly train the LEBM, generator and the data-space EBM to see whether the composed methods achieve higher performance. These methods are not direct competitors, but can be complementary to each other to further improve previous methods. In our initial submission here, we prefer to keep our model and learning method clean and simple, without involving extra networks and learned computations. We are happy to explore these directions in future works.
> > >
> > > - > One may doubt whether the proposed method can work or not when using a deeper architecture.
> > >
> > >     We would like to point out that in sec. 4.1 L215-228 and tab. 1 and fig. 2 CelebA-HQ columns we have explicitly provided the results on higher dimension data (256x256) to show that our method can scale up with deeper architecture (supp. C tab. 2).
> > >
> > >     To further verify the effectiveness of this method on more advanced architecture, in sec. 4.1 *GAN inversion* paragraph L229-254 we have provided further results using the StyleGAN network as the generator.
> > >
> > >     Therefore, we believe we have provided positive evidence that our method can effectively work with deeper architectures to scale up.
> > >
> > > - > Suggest that the authors show results with a bit deeper advanced network, which is not difficult for experiments.
> > >
> > >     Thank you for the constructive suggestion. Due to the time limit, we were only able to experiment on slightly deeper generators and report very preliminary results on CIFAR-10. Here we only add `conv3x3` layers to the original one; `+N` shows the number of additional layers and `w/ res` means adding residual connection to the conv layers. Please note that these models have not reached their best performances, but have already shown some initial improvements over the original model.
> > >
> > >     ||+0|+1|+2|+4 w/ res|
> > >     |:--:|:--:|:--:|:--:|:--:|
> > >     |FID|57.72|57.63|55.50|54.59|
> > >
> > > We hope our response helps to address your concerns. Please feel free to let us know if you have any additional questions.

---

### Official Review · Reviewer_NcyD · 2023-07-13

**Soundness:** 4 excellent
**Presentation:** 4 excellent
**Contribution:** 4 excellent
**Rating:** 8
**Confidence:** 3

**Summary:**

The authors propose DAMC, an amortization of MCMC sampling, via a scheme based on diffusion models, as an alternative to pure MCMC sampling, which usually suffers from either long mixing time or from being short and biased,for  priors and posteriors in energy based models. The method is theoretically sound, and experimentally argued for.

**Strengths:**

The theoretical analysis is sound. The algorithm is clear. The experiments are convincing.

**Weaknesses:**

N/A

**Questions:**

Are there some constraints that the latent space needs to satisfy for the proposed method to have an advantage over the vanilla MCMC counterpart ?

**Limitations:**

I see None for now.

---

> ### Author Rebuttal · Authors · 2023-08-09
>
> ### Thank you for your insightful comments
> We sincerely thank you for your kind words and thoughtful comments! Below, we provide point-to-point responses to hopefully address the concerns you have.
>
> - > Are there some constraints that the latent space needs to satisfy for the proposed method to have an advantage over the vanilla MCMC counterpart?
>     - Thank you for this very insightful input! For now, we see no obvious constraints on the latent space for the proposed method to outperform the vanilla short-run MCMC counterpart. We observed that our method shows a clearer advantage over the vanilla short-run MCMC method when the target distribution is highly multimodal or generally hard for short-run MCMC to fully explore.
>     - We also see no obvious issues for now on generalizing our method to learning amortized samplers for (unnormalized) densities or distributions other than learning energy-based models (including those arise in molecule dynamics and simulated annealing optimization). We are excited about these problem as a direction for future work.
>
> Thank you again for providing constructive and thoughtful feedback on our submission. If you have any additional questions/comments/concerns, please feel free to let us know here.

---

> > ### Comment · Reviewer_NcyD · 2023-08-14
> >
> > Thank you for your answers !

---

### Official Review · Reviewer_ga25 · 2023-07-16

**Soundness:** 3 good
**Presentation:** 2 fair
**Contribution:** 3 good
**Rating:** 6
**Confidence:** 3

**Summary:**

This paper proposed a diffusion-based amortised method to address the short-run MCMC samplers issues in the latent-space energy-based models. One interesting part is that it interleaves the distill T-steps of Langevin dynamics and KL divergence minimisation to sample from the target distribution $\pi$. Regarding the choice of amortised sampler parameters $\phi$, it uses the gradient of DDPM objective function to optimise them. The experimental results has verified the effectiveness of the DAMC.

**Strengths:**

The paper has the following strengths:

**1**, the usages of KL minimization and amortised MCMC transition is a well defined framework. Also, the usages of DDPM gradient for the long-run MCMC sampling in LEBMs is interesting.

**2**, the experimental evaluation is sufficient and clearly verifies the advantages of DAMC.

**Weaknesses:**

Regarding the weakness, I think the paper may need another round of polish. Some notations are not well defined before use and some introductory statement is not consistent. For example:

**1**, p_{uncond} may need more explanation, rather than a short word in the input of Algorithm.
**2**, $z_+^{(i)}, z_-^{(i)}$ are a bit confusing.
**3**, it may be a better idea if the notations of section 2.1 and 2.2 are consistent

**Questions:**

Sorry I am not an expert in this topic. I do not have questions for the authors.

**Limitations:**

I believe the authors has addressed the limitations.

---

> ### Author Rebuttal · Authors · 2023-08-09
>
> ### Thank you for your constructive comments
> We sincerely thank you for your kind words and thoughtful comments! Below, we provide point-to-point responses to hopefully address the concerns you have.
>
> - > p_{uncond} may need more explanation, rather than a short word in the input of Algorithm. $z_{+}$ and $z_{-}$ are a bit confusing.
>     - Thank you for pointing this out. We have introduced the meaning of p_{uncond} in sec 3.3 L175-180. $z_{+}$ and $z_{-}$ denotes the posterior and prior samples respectively.
>     - We agree that more explicit explanation of the hyperparameter and notations could make this draft more readable. We will make sure to revise our paper accordingly.
>
> - > It may be a better idea if the notations of section 2.1 and 2.2 are consistent
>     - Thank you for bringing this matter to our attention! We choose these notations to make sure that they are consistent with those used in Fig 1. We will revise these notations according to your kind suggestion.
>
> Thank you again for providing constructive and detailed feedback on our submission! If you have any additional questions/comments/concerns, please feel free to let us know here.

---

### Official Review · Reviewer_x1ko · 2023-07-16

**Soundness:** 3 good
**Presentation:** 3 good
**Contribution:** 2 fair
**Rating:** 6
**Confidence:** 3

**Summary:**

The paper proposes a diffusion-based amortized MCMC method for sampling the prior and posterior in latent space energy-based models. The paper provides some theoretical evidence using directly the result from Li et al., 2017. The paper shows the effectiveness of the proposed method throughout an extensive campaign on a variety of tasks such as image generation and construction, anomaly detection.

**Strengths:**

- This paper aims at tackling an important task in deep latent variable models, which is to improve energy-based prior model.
- The paper is well-motivated and is well-written.
- The experiments on image datasets are extensive and diverse.

**Weaknesses:**

- The idea of amortizing the short-run MCMC sampling of the prior and posterior distributions is already proposed in the original work of Pang et al., 2020. This paper aims to improve this amortization using a diffusion model. In addition, the theoretical contribution is weak, as the convergence property is already shown by Li et al., 2017.
- Although the authors claim that the theoretical evidence that the learned amortization of MCMC is a valid long-run MCMC sampler, this result is taken directly from Li et al., 2017. As such, the authors should show that the proposed amortization scheme can converge to the true distributions, at least via toy examples. For example, the authors could use examples from this work [1].
- The improvement is expectable as the authors use a very powerful diffusion model to amortize the prior and posterior. However, this increases the computational costs, as the authors discussed at the end of Section 4.1.
- As, again, the authors employ a power diffusion model in latent space. To be fair, the authors should consider the baselines of using diffusion models in the latent space of latent variable models; for example [2], at the very least.

[1] Taniguchi et al., Langevin Autoencoders for Learning Deep Latent Variable Models. NeurIPS 2022.

[2] Pandey et al., DiffuseVAE: Efficient, Controllable and High-Fidelity Generation from Low-Dimensional Latents. TMLR 2022.



**Questions:**

- There are a lot of hyper-parameters in the proposed methods. How do the authors choose $T$?
- It would be great if the authors could ablate the amortization gap of the proposed method by considering different capacities of the $q_{\phi}$

**Limitations:**

Yes

---

> ### Author Rebuttal · Authors · 2023-08-09
>
> ### Thank you for your detailed and insightful comments
> We sincerely thank you for your time and constructive comments! Below, we provide point-to-point replies to your comments that hopefully would address the concerns you have.
>
> - > The idea of amortizing the short-run prior and posterior MCMC sampling is already proposed in [1].
>     - As mentioned in the 2.6 and A.11 sections of [1], they propose to amortize the posterior sampling process with the
>       network $q_{\phi}(z|x)$ following the variational learning (VI) scheme introduced by [2]. However, the Gaussian (or other tractable density) assumption
>       of the posterior distribution made in this scheme could greatly limit the expressivity of $q_{\phi}(z|x)$ [3,4]. In this work, we use a conditional diffusion model to address this issue. As shown in tab. 1, we have compared our model with the NCP-VAE baseline, which indeed trains the inference network through VI to learn the energy-based prior. We can see that our method significantly outperforms the NCP-VAE. On the CIFAR-10 dataset, the FID and MSE scores are 78.06 vs. 57.72, and 0.054 vs. 0.015.
>     - In the A.11 section, [1] mentioned that one can learn a synthesis network $q_{\psi}(z)$ for prior sampling, but did not provide a concrete solution for this problem. Model design for amortizing the prior sampling requires non-trivial
>       extra efforts, since the latent space is ever-changing during training; delicate designs are needed to balance the learning of this amortized model and other jointly learned models for stable training. In this work, we follow the method of [5] to train a single neural network to parameterize both the prior and posterior sampling models to deal with this problem.
>
> - > The authors should show that the proposed this amortization scheme can converge to the true distributions, at least via toy examples in [4].
>     - Thank you for pointing us to this very interesting example! We have implemented the neural likelihood examples following the same set-up mentioned in sec. 5.1 and B.1 in [6]. We choose to use a more complex prior distribution, i.e., 2-arm pinwheel-shaped prior distribution instead of a standard normal one to make sure that the true posterior distributions are multimodal. We have attached the visualization of the convergence of the learned posterior distributions by our model to the ground-truth ones (please see the global response). We can see that our model could faithfully reproduce the ground-truth distributions. We will add these results in revision and make explicit connections to [4].
>
> - > The authors use a very powerful diffusion model to amortize the prior and posterior. However, this increases the computational costs.
>     - We would like to point out that we actually used a very light-weight MLP-based diffusion for all the experiments in this paper, since our models are in the low-dimensional latent space. As mentioned in sec 4.1 L257-258 in the main text and G.1 in the appendix, the number of parameters in the diffusion model is only around 10% of those in the generator.
>     - Consequently, the computational costs induced by this model is very marginal. To be specific, as mentioned in sec 4.1 L258-261, with the batch size of 64, our method takes ~0.3s for prior sampling, while 100 steps of short-run LD with LEBM takes 0.2s; posterior sampling with the proposed method takes ~1.0s, while posterior LD sampling takes ~8.0s because it requires back-propagations through a much heavier generator network.
>
> - > Should consider the baselines of using diffusion models in the latent space of latent variable models; for example [6].
>     - Thank you for pointing us to this very interesting work! However, we notice that DiffuseVAE actually trains a diffusion model in the pixel space to refine the output of a VAE model. This is essentially different from our set-up, where we learn the models in the latent space of generative models. We will dicuss the key differences between DiffuseVAE and our work in revision.
>     - We have compared our method with directly training diffusion models in the latent space. In the NALR-LEBM column in tab. 4 in sec 4.3 and L292-300, we compared our method with learning a diffusion model in the pre-trained energy-based latent space on CIFAR-10 dataset. We further add experiments of learning this model in the pre-trained VAE and ABP model latent spaces summarized below. We can see that our method greatly outperforms these baselines using the same network archs.
> ||VAE|ABP|NALR|Ours
> |:--:|:--:|:--:|:--:|:--:|
> |FID|102.54|69.93|64.38|**57.52**|
> |MSE|0.036|0.017|0.016|**0.015**|
>
> - > How do the authors choose $T$? Would be great to consider different capacities of the $q_{\phi}$.
>     - Thank you for bringing this matter to our attention! For prior sampling, we follow the set-up of [1] for fair comparison. We further add experiments on CIFAR-10 for different posterior steps and capacities of $q_{\phi}$ summarized below. We used $T=30$ for training in this paper. We can see that larger $T$ and larger $q_{\phi}$ brings very marginal improvement.
> ||T=10|T=30|T=50|
> |:--:|:--:|:--:|:--:|
> |FID|74.20|**57.72**| 57.03|
> |MSE|0.016|**0.015**| 0.015|
>     - Here $f$ stands for the factor for the $q_{\phi}$ capacity, e.g., $f=2$ means 2x the size of the original model.
> ||f=1/4 |f=1/2 |f=1|f=2|f=4
> |:--:|:--:|:--:|:--:|:--:|:--:|
> |FID|116.28| 80.28|**57.52**|57.82|57.56|
> |MSE|0.017| 0.016|**0.015**|0.015|0.015|
>
> Thank you again for providing constructive and detailed feedback on our submission. If you have any additional questions/comments/concerns, please feel free to let us know here. **Otherwise, we would appreciate it if you would consider raising your rating of this submission**.
>
> [1] Pang et al. LEBM. NeurIPS 2020.
>
> [2] Kingma et al. VAE. 2013
>
> [3] Rajesh et al. HVAE. ICML 2016.
>
> [4] Taniguchi et al. LAE. NeurIPS 2022.
>
> [5] Ho et al. Classifier-Free Guidance. NeurIPS 2021 Workshop
>
> [6] Pandey et al., DiffuseVAE. TMLR 2022

---

> > ### Comment · Reviewer_x1ko · 2023-08-10
> > **Post Rebuttal Reply**
> >
> > I thank the authors for their response and the effort they put in resolving my concerns. The new results are very encouraging.
> >
> > Might I kindly ask the authors to provide the code for toy examples? This would enable both me and other reviewers can quickly validate the proposed method and the results.

---

> > > ### Author Response · Authors · 2023-08-11
> > > **Anonymous Code Submission**
> > >
> > > We thank the reviewer for your prompt reply! We are pleased to hear that our new results are very encouraging. We have asked our AC for details about anonymous code submission, since we are only allowed to submit the code link to AC according to the instructions this year. We have prepared the code and will submit the link immediately once permitted. Here we provide more details about our code submission to help you reproduce and validate our results.
> > > - Environment Specification
> > >     - We use Pytorch to train our models. We did not use other third-party python packages for implementation. Please feel free to let us know if there are any dependency issues. We will be more than happy to help.
> > >     - Version of packages used in the toy example experiments:
> > >         - Python == 3.9.2
> > >         - Pytorch == 1.10.0
> > >         - numpy == 1.21.2
> > >         - matplotlib == 3.3.4
> > > - Code Structure
> > >
> > >     We have added comments to most functions and code blocks in our code. The code files should have the following structure:
> > >     ```
> > >     toy_code/
> > >         src/
> > >             diffusion_helper_func.py
> > >             diffusion_net.py
> > >         toy_example.py
> > >     ```
> > >     - `toy_example.py`is the main file. It includes the training algorithm, data generation pipeline and visualization functions.
> > >     - `src/diffusion_net.py` contains the detailed network structure of the diffusion network.
> > >     - `src/diffusion_helper_func.py` contains helper function for implementing the denoising diffusion process.
> > > - How to Run
> > >
> > >     To train the model on the toy example, you can simply run the following command in the `toy_example` folder.
> > >     ```
> > >     CUDA_VISIBLE_DEVICES=<DEVICE_ID> python toy_example.py --seed <RANDOM_SEED_TO_SPECIFY>
> > >     ```
> > >     For example,
> > >     ```
> > >     CUDA_VISIBLE_DEVICES=0 python toy_example.py --seed 0
> > >     ```
> > >     - Here `--seed` argument specifies the random seed, which basically decides the ground-truth posterior distribution. The script will automatically generate a `logs/toy/<TIMESTAMP>` folder in the `toy_code` folder, where `<TIMESTAMP>` indicates the time you started this training process.
> > >
> > >     - There will be two automatically created additional folders in `logs/toy/<TIMESTAMP>` once running the script: i) `ckpt` which saves the trained weights and ii) `viz` folder that saves the visualization of ground-truth and learned posterior distributions. The image file names are `<ITERATION>_lang_post_Q` and `<ITERATION>_lang_post_gt`, which indicates visualization of the learned distribution and the ground-truth distribution respectively. These visualization results are saved every 100 iterations.
> > > - Important Tips about Training
> > >     - For most random seeds, we observed that our learned sampler could achieve decent approximation of the ground-truth posterior distributions obtained by long-run langevin dynamics within 300-3000 training iterations. This would take from several minutes to an hour or so on a NVIDIA RTX A6000 GPU. The training process takes ~2GB GPU memory. It is possible that there are some extreme cases where longer training iterations are needed to produce decent results.
> > >     - For some random seeds, the default 1000-step langevin dynamics for sampling ground-truth posterior distribution might not converge. You may consider using 2000 or more steps by modifying the `g_l_steps` argument in the `sample_langevin_post_z` function at L277 in the `toy_example.py`. One possible sign is that the `g_loss (avg) Q` (reconstruction error obtained by learned posterior samples) is significantly lower than `g_loss (avg) L` (reconstruction error obtained by langevin dynamics samples).
> > >
> > > Finally, we kindly ask the reviewer to not distribute our code, since we have not officially published our work. Thank you again for your time for evaluating our work, please feel free to let us know if you have any additional questions/comments/concerns.

---

> > > ### Author Response · Authors · 2023-08-12
> > > **Anonymous Code Submitted**
> > >
> > > Thank you for your patience. We have submitted the anonymous code for reproducing the results for the toy example to our AC. Please feel free to let us know if you have any problems with the code.

---

> > > > ### Comment · Reviewer_x1ko · 2023-08-17
> > > > **Response from Reviewer x1ko**
> > > >
> > > > Thank the authors for providing the code.
> > > >
> > > > > For some random seeds, the default 1000-step langevin dynamics for sampling ground-truth posterior distribution might not converge. You may consider using 2000 or more steps by modifying the g_l_steps argument in the sample_langevin_post_z function at L277 in the toy_example.py. One possible sign is that the g_loss (avg) Q (reconstruction error obtained by learned posterior samples) is significantly lower than g_loss (avg) L (reconstruction error obtained by langevin dynamics samples).
> > > >
> > > > I encourage the authors to discuss the sensitivity of hyperparameters such as this in the paper.
> > > >
> > > > I've checked the code. While I could not achieve near-perfect approximations of the target distributions as presented in the rebuttal, the approximations I obtained are acceptable. If the authors cherry-picked results, I encourage the authors to showcase not only favorable outcomes but also include instances that demonstrate less favorable results.
> > > >
> > > > I strongly recommend the authors incorporate the new results in the rebuttal into the next version of the paper as I believe they will strengthen the paper.
> > > >
> > > > In light of the new results and the responses from the authors, I will increase my score to 6.

---

> > > > > ### Author Response · Authors · 2023-08-20
> > > > > **Thank you!**
> > > > >
> > > > > Thank you for your kind response and constructive feedback!
> > > > >
> > > > > For the toy example, we believe it is largely because of the steps of Langevin dynamics for sampling the ground-truth posterior distribution mentioned in our previous response. We will add more explicit illustrations in the revision of our paper for clearer guidance for obtaining our results.
> > > > >
> > > > > We would like to thank the reviewer again for the detailed and thoughtful comments. We will make sure to add these new results to the next version of this paper to further improve the quality of this work.

---

### Author Rebuttal · Authors · 2023-08-09

### Summary of our response
We thank the reviewers for their insightful and constructive comments and careful reviews of our paper! We appreciate that the reviewers consider our submission "well-written and well-motivated", "clearly stated", "new" and "interesting" and provide "convincing", "diverse" and "extensive" experiment results. We have provided point-to-point replies to your comments that hopefully would address the remaining concerns you have. We summarize our response as follows:
- Clarification
    - Key differences between our method and the ideas mentioned in [1, 2] and [3, 4]. (Response for reviewer x1ko and hz9t)
    - Discussion for advantages of energy-based parametrization over epsilon-parameterization (Response for reviewer rkuE)
- Further Experiments
    - Toy example from [5] as a proof-of-concept to show that the proposed amortization scheme can converge to the true distributions. Please see the attached PDF file for the additional results. (Response for reviewer x1ko)
    - Adding more baselines including training DDPMs in the latent spaces of VAE and ABP models using the current network architectures. (Response for reviewer x1ko)
    - Adding ablation studies for steps of Langevin Dynamics $T$ and the model capacity of the amortizer $q_{\phi}$. (Response for reviewer x1ko)
    - Adding quantitative results for longer-run chain analysis. (Response for reviewer hz9t).
- Including missing references pointed out by reviewers.
- Polishing notations and writing in general as suggested by reviewers.

Thank you again for providing constructive and detailed feedback on our submission. If you have any additional questions/comments/concerns, please feel free to let us know.

[1] Pang et al. Learning Latent Space Energy-Based Prior Model. NeurIPS 2020.

[2] Pandey et al. DiffuseVAE: Efficient, Controllable and High-Fidelity Generation from Low-Dimensional Latents. TMLR 2022.

[3] Xie et al. Cooperative Training of Descriptor and Generator Networks. TPAMI 2018.

[4] Song and Ou. Learning Neural Random Fields with Inclusive Auxiliary Generators. arXiv:1806.00271.

[5] Taniguchi et al. Langevin Autoencoders for Learning Deep Latent Variable Models. NeurIPS 2022.

---

### Decision · Program_Chairs · 2023-09-21

**Decision:**

Accept (poster)

**Comment:**

The paper offers an innovative approach to address the sampling issue found when learning latent space Energy-Based Models (EBMs). By using a diffusion-based amortized Markov Chain Monte Carlo (MCMC) method, the paper aims to enhance the sampling of the prior and posterior in such models.

The paper provides extensive simulations and convincingly shows the effectiveness of the proposed method, comparing favourable to other leading techniques. Furthermore, the application of Denoising Diffusion Probabilistic Models (DDPM) gradient for long-run MCMC sampling in Latent Energy-Based Models (LEBMs) is an interesting choice, indicating the integrated use of established methodologies for more efficient and effective sampling.

Two reviewers were leaning towards a rejection -- after careful examination of these particular reviews, it was concluded that the objections raised were based on an extremely superficial reading of the manuscript. It is demonstrated by the shortness of these particular reviews (and their content!), the low confidence of the reviewers, as well as the discussions with the authors during the rebuttal period.

Overall, the paper is recommended for acceptance as it successfully tackles a critical challenge in the domain of deep latent variable models, namely the design of more efficient methodologies for learning  energy-based prior models. The proposed is a promising contribution to the field of generative modeling.